# A STATISTICAL FRAMEWORK FOR RANKING LLM-BASED CHATBOTS

**Siavash Ameli**
ICSI and Department of Statistics
University of California, Berkeley
sameli@berkeley.edu

**Siyuan Zhuang**
Department of Computer Science
University of California, Berkeley
siyuan_zhuang@berkeley.edu

**Ion Stoica**
Department of Computer Science
University of California, Berkeley
istoica@cs.berkeley.edu

**Michael W. Mahoney**
ICSI, LBNL, and Department of Statistics
University of California, Berkeley
mmahoney@stat.berkeley.edu

## ABSTRACT

Large language models (LLMs) have transformed natural language processing, with frameworks like Chatbot Arena providing pioneering platforms for evaluating these models. By facilitating millions of pairwise comparisons based on human judgments, Chatbot Arena has become a cornerstone in LLM evaluation, offering rich datasets for ranking models in open-ended conversational tasks. Building upon this foundation, we propose a statistical framework that incorporates key advancements to address specific challenges in pairwise comparison analysis. First, we introduce a factored tie model that enhances the ability to handle ties—an integral aspect of human-judged comparisons—significantly improving the model's fit to observed data. Second, we extend the framework to model covariance between competitors, enabling deeper insights into performance relationships and facilitating intuitive groupings into performance tiers. Third, we resolve optimization challenges arising from parameter non-uniqueness by introducing novel constraints, ensuring stable and interpretable parameter estimation. Through rigorous evaluation and extensive experimentation, our framework demonstrates substantial improvements over existing methods in modeling pairwise comparison data. To support reproducibility and practical adoption, we release `leaderbot`, an open-source Python package implementing our models and analyses.

## 1 INTRODUCTION

The rapid advancement of large language models (LLMs) has transformed natural language processing, enabling breakthroughs across diverse tasks. As these models evolve, the need for effective evaluation methods becomes crucial for fostering innovation and ensuring that LLMs align with human preferences. Traditional benchmarks, such as MMLU (Hendrycks et al., 2021) and HumanEval (Chen et al., 2021), play an important role in assessing specific capabilities of LLMs. However, they often fall short in capturing the nuanced, real-world interactions characteristic of open-ended conversational tasks, particularly those seen in chatbot applications.

To address this gap, crowdsourced evaluation platforms have emerged, with Chatbot Arena (Chiang et al., 2024; Zheng et al., 2023) standing out as a pioneering framework. By facilitating millions of pairwise comparisons between LLMs based on human judgments, Chatbot Arena has become one of the largest and most credible datasets (Zheng et al., 2024) for chatbot evaluation. Its design more closely reflects the open-ended nature of chatbot usage, providing unparalleled diversity and robustness in assessing model performance. In its first year, the platform amassed over two million votes across more than 150 state-of-the-art models, gaining adoption by leading institutions such as OpenAI, Google, and Hugging Face. This unparalleled scale and impact have solidified Chatbot Arena as a cornerstone in the LLM evaluation ecosystem.

The ranking methodology employed in Chatbot Arena relies on the Elo rating system (Zermelo, 1929; Bradley & Terry, 1952), which is well-suited for competitive settings with clear win-loss outcomes. However, the Elo system does not account for ties—a notable portion of human-judged comparisons—or for modeling deeper relationships between competitors, such as correlations in performance. Addressing these aspects presents an opportunity to build upon the success of Chatbot Arena, enhancing its analytical capabilities while retaining its foundational strengths.

In this paper, we propose a statistical framework that extends the foundational Elo-based approach employed in Chatbot Arena. Our contributions include:

1. **Incorporating ties:** Ties, where two competitors are judged to perform equally, are a key feature of pairwise comparisons. To model ties, we integrated well-established methods by Rao & Kupper (1967) and Davidson (1970), which define tie probabilities based on axiomatic assumptions. These methods have been applied in various contexts, from sports analytics to marketing and beyond (see Appendix A). While these frameworks offer a solid foundation for paired comparisons, applying them directly to the Chatbot Arena dataset, with its extensive pairwise comparisons, highlighted the need for further adaptation: these models yielded errors exceeding 10% when applied to tie outcomes.

   To address these challenges, we propose a novel factored tie model, which generalizes these frameworks by uncovering latent structures in tie patterns across pairs of competitors. This factor analysis substantially improves model fit, reducing errors in fitting tie data by two orders of magnitude. Notably, this enhancement also extends to win/loss predictions, achieving comparable improvements. Details of tie modeling, including a background and our generalizations, are provided in Section 2.

2. **Incorporating covariance:** We extend paired comparison models by introducing Thurstonian representations to capture covariance structures between competitors (Section 2.4), enabling deeper exploration of relationships beyond rankings, such as correlations in performance. Building on our theoretical analysis of covariance's structural non-uniqueness and equivalence classes (Appendices B and C), we show that derived metrics from covariance, such as dissimilarity metrics, are unique and provide interpretable insights. These metrics enable visualization techniques to uncover latent patterns, such as performance trends, and clustering techniques to group competitors into performance tiers based on relative strengths.

3. **Addressing optimization challenges through constraints:** Paired comparison models often exhibit parameter non-uniqueness due to structural symmetries in the model, resulting in equivalent parameter configurations that leave the likelihood invariant. These symmetries lead to valid but indistinguishable solutions, compromising convergence stability during likelihood optimization. To resolve these issues, we introduce novel constraints that regularize the parameter space, ensuring stable and interpretable parameter estimation (Section 2.5 and Appendix C.4).

In addition to developing our framework, we conducted comprehensive evaluations and analyses to validate its performance and interpret its results. Empirical evaluations (Section 3) demonstrate the model's fit to observed data, supported by extensive experimentation (Appendix D) on model selection, goodness-of-fit, and prediction metrics. Detailed inference analyses (Section 4) further explore competitor relationships, offering visual insights through clustering and ranking similarity analyses (Appendices C and E). We also examine the relationship between LLM characteristics and their performance scores in Appendix F. Furthermore, Appendix A highlights the framework's potential applicability beyond ranking and chatbot evaluations to other areas of machine learning.

To support reproducibility and broader adoption, we provide `leaderbot`, an open-source Python package implementing our statistical framework with tools for data processing, model fitting, and visualization. This ensures that all results in this paper are fully reproducible (Appendix G).

## 2 STATISTICAL MODEL

### 2.1 PROBLEM STATEMENT

Consider a paired-comparison experiment involving $m \geq 2$ competitors (here, LLM chatbots), indexed by the set $V := \{1, \ldots, m\}$. Let $E \subseteq \{\{i, j\} \mid i, j \in V\}$ denote the set of unordered pairs of competitors that have been compared in the experiment. We assume the graph $\mathcal{G}(V, E)$ is connected.

We define the $m \times m$ matrix $\mathbf{W} = [w_{ij}]$, where $w_{ij}$ represents the frequency with which competitor $i$ wins against competitor $j$, and $w_{ji}$ represents the frequency with which $i$ loses to $j$. Similarly, we define the symmetric $m \times m$ matrix $\mathbf{T} = [t_{ij}]$, where $t_{ij}$ denotes the frequency of ties between competitors $i$ and $j$, with $t_{ij} = t_{ji}$. We set $w_{ij} = w_{ji} = t_{ij} = 0$ whenever $\{i, j\} \notin E$ to reflect the absence of comparisons between competitors $i$ and $j$. The total number of comparisons between competitors $i$ and $j$ is denoted by $n_{ij}$, where $n_{ij} = w_{ij} + w_{ji} + t_{ij}$. The triple $(\mathcal{G}, \mathbf{W}, \mathbf{T})$ constitutes the input data for our problem.

Our objective is to rank the competitors based on their performance in the overall comparisons. To formalize this in a probabilistic framework, we define $P(i \succ j \mid \{i, j\})$ as the probability that competitor $i$ wins against competitor $j$, and $P(i \sim j \mid \{i, j\})$ as the probability that $i$ and $j$ tie. For notational simplicity, we often denote these probabilities by $P_{i \succ j}$ and $P_{i \sim j}$, respectively.

A broad class of parametric models (which we will discuss in detail later) assumes the existence of a *score* array $\boldsymbol{x} = (x_1, \ldots, x_m) \in \mathbb{R}^m$, which defines the ranking. Specifically, the ranking is inferred by a bijection from $V$ to itself that orders the scores $x_i$, such that $x_i > x_j$ implies $i$ is ranked higher than $j$, denoted by the binary relation $i \succ j$.

The score vector $\boldsymbol{x}$ forms part of the model's parameters, denoted by $\boldsymbol{\theta}$, which also includes other parameters governing the probability of each outcome. A common approach for estimating these parameters is the maximum likelihood method. The likelihood function $\mathcal{L}(\boldsymbol{\theta} \mid \mathcal{G}, \mathbf{W}, \mathbf{T})$ is defined as the product of multinomial distributions for each compared pair $\{i, j\} \in E$, given by

$$\mathcal{L}(\boldsymbol{\theta} \mid \mathcal{G}, \mathbf{W}, \mathbf{T}) = \prod_{\{i,j\} \in E} \frac{n_{ij}!}{w_{ij}! w_{ji}! t_{ij}!} P_{i \succ j}^{w_{ij}}(\boldsymbol{\theta}) P_{i \prec j}^{w_{ji}}(\boldsymbol{\theta}) P_{i \sim j}^{t_{ij}}(\boldsymbol{\theta}). \tag{1}$$

The parameter estimate $\boldsymbol{\theta}^*$ is then obtained by maximizing the log-likelihood function $\ell(\boldsymbol{\theta}) := \log \mathcal{L}(\boldsymbol{\theta})$, i.e., $\boldsymbol{\theta}^* = \mathrm{argmax}_{\boldsymbol{\theta}} \, \ell(\boldsymbol{\theta})$.

## 2.2 PROBABILISTIC MODELS

A parametric model for the above probabilities must satisfy two fundamental axioms. First, by the law of total probability, we have $P_{i \succ j} + P_{i \prec j} + P_{i \sim j} = 1$. Second, the model should respect the concept of transitivity in ranking, though in a probabilistic setting. Rather than standard transitivity, where $i \succ j$ and $j \succ k$ imply $i \succ k$, we adopt the principle of *stochastic transitivity* (see, e.g., Shah et al. (2017); Shah & Wainwright (2018)).

In particular, we are interested in *strong stochastic transitivity*, which states that if $P_{i \succ j} \geq \frac{1}{2}$ and $P_{j \succ k} \geq \frac{1}{2}$, then $P_{i \succ k} \geq \max\{P_{i \succ j}, P_{j \succ k}\}$. A key sub-class of strong stochastic transitivity, and the focus of this work, is *linear stochastic transitivity*. This property is characterized by the existence of an increasing *comparison function* $F : \mathbb{R} \to [0, 1]$ and a *merit function* $\zeta : \mathbb{R} \to \mathbb{R}$, such that $P_{i \succ j} = F(\zeta(x_i) - \zeta(x_j))$.

In the following subsections, we describe several common models of paired comparison that satisfy these properties.

### 2.2.1 BRADLEY-TERRY MODEL

One of the most widely used models for paired comparison was first introduced by Zermelo (1929) and later rediscovered by Bradley & Terry (1952), leading to the model being named after them. The Bradley-Terry model forms the basis of the well-known Elo rating system, which is extensively used by the World Chess Federation. In this model, the probabilities of win and loss are given by

$$P(i \succ j \mid \{i, j\}) := \frac{\pi_i}{\pi_i + \pi_j} \quad \text{and} \quad P(i \prec j \mid \{i, j\}) := \frac{\pi_j}{\pi_i + \pi_j}, \tag{2}$$

where $\pi_i := e^{x_i}$. This model assumes that $x_i - x_j$ follows a logistic distribution, as shown by

$$P(x_i - x_j > 0) = \frac{1}{1 + e^{-(x_i - x_j)}}. \tag{3}$$

Variants of this model, such as the one proposed by Glenn & David (1960), assume a standard normal distribution instead of the logistic. However, the logistic distribution is typically preferred in paired comparison settings due to its heavier tails, which provide better fit for real-world data, and its computational advantages and tractability (Böckenholt, 2001).

### 2.2.2 MODELS WITH TIES

The Bradley-Terry (BT) model does not account for ties (i.e., $P_{i\sim j} = 0$), making it more suited for balanced paired comparisons, such as zero-sum games. However, in our application of ranking LLM chatbot agents, ties frequently occur, which poses challenges for applying the BT model directly.

Previous work, such as Chiang et al. (2024), addressed this by treating a tie as halfway between a win and a loss, modifying the outcome matrix as $\mathbf{W} \leftarrow \mathbf{W} + \frac{1}{2}\mathbf{T}$. While this approach provides a straightforward way to handle ties, other extensions of the BT model incorporate ties through axiomatic frameworks, which we explore in the following sections.

One such generalization is the model of Rao & Kupper (1967), which modifies the logistic distribution to account for ties. The resulting probabilities for win, loss, and tie are given by

$$P(i \succ j \,|\, \{i,j\}) = P(x_i - x_j > \eta) := \frac{\pi_i}{\pi_i + \nu\pi_j}, \tag{4a}$$

$$P(i \prec j \,|\, \{i,j\}) = P(x_i - x_j < -\eta) := \frac{\pi_j}{\nu\pi_i + \pi_j}, \tag{4b}$$

$$P(i \sim j \,|\, \{i,j\}) = P(|x_i - x_j| < \eta) := \frac{\pi_i\pi_j(\nu^2 - 1)}{(\pi_i + \nu\pi_j)(\nu\pi_i + \pi_j)}, \tag{4c}$$

where $\nu := e^\eta \geq 1$ and $\eta \geq 0$ is a threshold parameter to be optimized. In this model, if the difference between competitors' scores is less than the threshold $\eta$, the judge is unable to distinguish between competitors $i$ and $j$, resulting in a tie. Setting $\eta = 0$ reduces this model to the standard BT model.

Another extension of the BT model, introduced by Davidson (1970) and based on the axiom of choice of Luce (1959), models ties differently:

$$P(i \succ j \,|\, \{i,j\}) := \frac{\pi_i}{\pi_i + \pi_j + \nu\sqrt{\pi_i\pi_j}}, \tag{5a}$$

$$P(i \prec j \,|\, \{i,j\}) := \frac{\pi_j}{\pi_i + \pi_j + \nu\sqrt{\pi_i\pi_j}}, \tag{5b}$$

$$P(i \sim j \,|\, \{i,j\}) := \frac{\nu\sqrt{\pi_i\pi_j}}{\pi_i + \pi_j + \nu\sqrt{\pi_i\pi_j}}, \tag{5c}$$

where $\nu := e^\eta$ and $\eta \in \mathbb{R}$ is a threshold parameter for tie to be optimized. Here, setting $\eta = -\infty$ reduces this model to the BT model. Note that in both the Rao-Kupper and Davidson models, the probability of a tie increases as the difference in scores decreases.

### 2.3 A GENERALIZATION FOR MODELS WITH TIES

The original Rao-Kupper and Davidson models each employ a single parameter for ties, $\nu$. However, as our numerical results will demonstrate, one tie parameter is insufficient to capture the complexity of ties across all pairs of competitors. To address this, we propose a generalization of these models by incorporating additional parameters, which, to our knowledge, is a novel extension.

In this generalized model, the tie parameter $\nu = e^\eta$ is replaced with a pair-specific parameter $\nu_{ij} = e^{\eta_{ij}}$, where $i, j \in V$, subject to the symmetry condition $\nu_{ij} = \nu_{ji}$. This modification introduces $|E|$ parameters, potentially leading to overfitting. To mitigate this, we propose a reduced model. Let the symmetric $m \times m$ matrix $\mathbf{H} = [\eta_{ij}]$ represent the pairwise tie parameters. Rather than treating all $\eta_{ij}$ as independent parameters, we introduce the following factor model to construct $\mathbf{H}$ by

$$\mathbf{H} := \begin{cases} \mathbf{G}\mathbf{\Phi}^\mathsf{T} + \mathbf{\Phi}\mathbf{G}^\mathsf{T}, & 0 < k_{\mathrm{tie}} \leq m, \\ \eta\mathbf{J}, & k_{\mathrm{tie}} = 0, \end{cases} \tag{6}$$

where $\mathbf{G} = [g_{ij}]$ is an $m \times k_{\mathrm{tie}}$ matrix of the new parameters $g_{ij}$, $\mathbf{\Phi} = [\phi_{ij}]$ is an $m \times k_{\mathrm{tie}}$ constant matrix of rank $k_{\mathrm{tie}} \leq m$ consisting of predefined basis column vectors that will be discussed below, and $\mathbf{J}$ is the $m \times m$ matrix of all ones. The rank of $\mathbf{H}$ is $\max(1, \min(2k_{\mathrm{tie}}, m))$ containing $\max(1, mk_{\mathrm{tie}})$ parameters; and thus choosing $k_{\mathrm{tie}}$ allows us to strike a balance between the goodness

of fit and the complexity of the model. Note that the case $k_{\text{tie}} = 0$ reverts to the original Rao-Kupper or Davidson models where a single parameter $\eta_{ij} = \eta$ is used.

To further motivate the construction in (6), the pairwise tie parameters $\eta_{ij}$ (when $k_{\text{tie}} \neq 0$) can be expressed as

$$\eta_{ij} = \boldsymbol{g}_i \cdot \boldsymbol{\phi}_j + \boldsymbol{g}_j \cdot \boldsymbol{\phi}_i, \tag{7}$$

where $\boldsymbol{g}_i \coloneqq (g_{i1}, \ldots, g_{ik_{\text{tie}}})$ and $\boldsymbol{\phi}_i \coloneqq (\phi_{i1}, \ldots, \phi_{ik_{\text{tie}}})$ are the $i$-th rows of $\mathbf{G}$ and $\boldsymbol{\Phi}$, respectively. This formulation emphasizes the *additive* nature of our proposed model, where contributions from competitors $i$ and $j$ are independently and symmetrically combined. While designing this generalization, we also considered adopting a *multiplicative* structure (e.g., $\mathbf{H} = \mathbf{A}\mathbf{A}^\mathsf{T}$), as is commonly used in factor analysis. However, we found it unsuitable for this application: low contributions from one competitor's tie threshold parameters would disproportionately suppress the pairwise tie parameter $\eta_{ij}$, leading to an ineffective model for ties. By contrast, our additive design reflects the cumulative nature of ties, which we found aligns naturally with the structure of pairwise comparisons and is particularly effective for modeling shared thresholds.

To ensure $\boldsymbol{\Phi}$ is of full rank, we design $\boldsymbol{\Phi}$ with orthogonal column vectors. This can be achieved, for instance, by orthogonalizing a randomly generated matrix using Gram-Schmidt orthogonalization. However, for reproducibility and to avoid using randomly generated matrices, we recommend constructing $\boldsymbol{\Phi}$ using discrete orthogonal polynomials with respect to a uniform weight over equally spaced points (Baik et al., 2007), which are inherently orthogonal. Such matrices can be generated, for instance, by discrete Legendre polynomials, the Hadamard transform (when $m$ is a power of 2), discrete Chebyshev polynomials (Corr et al., 2000), or the discrete cosine transform (DCT). For simplicity, we choose the discrete cosine transform of the fourth type (DCT-IV) basis, where the elements $\phi_{ij}$ are given by

$$\phi_{ij} = \sqrt{\frac{2}{m}} \cos\left(\frac{\pi}{4m}(2i-1)(2j-1)\right), \quad i = 1, \ldots, m, \quad \text{and} \quad j = 1, \ldots, k_{\text{tie}}. \tag{8}$$

## 2.4 INCORPORATING COVARIANCE USING THURSTONIAN REPRESENTATIONS

A fundamental approach to modeling paired comparisons was introduced by Thurstone (1927) through the laws of comparative judgment, laying the foundation for psychometric choice modeling from a statistical perspective. In this section, we build on Thurstone's multivariate discriminal process to incorporate covariance structures into the paired comparison models discussed earlier, enriching the modeling framework with covariance as an additional parameter.

Thurstonian models assume that the score variables $\boldsymbol{x}$ are stochastic processes defined by $\boldsymbol{x} = \boldsymbol{\mu} + \boldsymbol{\epsilon}$, where $\boldsymbol{\mu} \in \mathbb{R}^m$ is the deterministic component representing the mean of the process, and $\boldsymbol{\epsilon}$ is a zero-mean random component with covariance $\boldsymbol{\Sigma} = [\sigma_{ij}]$, where $\sigma_{ij} \coloneqq \text{cov}(\epsilon_i, \epsilon_j)$, often referred to as the *comparative dispersion*. In this model, $\boldsymbol{\mu}$ and $\boldsymbol{\Sigma}$ serve as the parameters, with $\boldsymbol{\mu}$ determining rankings by reflecting the expected performance of competitors.

We note that in this model, the difference between scores, which is central to the previously mentioned models, is also a stochastic process: $x_i - x_j = \mu_i - \mu_j + \epsilon_{ij}$. Here, $\epsilon_{ij}$ has variance $\mathbf{S} = [s_{ij}]$, defined as $s_{ij} \coloneqq \text{var}(\epsilon_{ij})$, and given by $s_{ij} = \sigma_{ii} + \sigma_{jj} - 2\sigma_{ij}$.[1] This is commonly referred to as the *discriminal dispersion* (Heiser & de Leeuw, 1981).

The original Thurstonian model assumes that $\boldsymbol{x}$ follows a joint normal distribution, meaning that $x_i - x_j$ also has a normal distribution. Let $x_{ij} \coloneqq x_i - x_j$, $\mu_{ij} \coloneqq \mu_i - \mu_j$, and $y_{ij} \coloneqq (x_{ij} - \mu_{ij})/\sqrt{s_{ij}}$. It can be shown that $P_{i \succ j} = P(x_{ij} > 0) = \Phi(z_{ij})$ (Maydeu-Olivares, 1999), where $\Phi$ is the cumulative standard normal distribution, and

$$z_{ij} \coloneqq \frac{\mu_i - \mu_j}{\sqrt{s_{ij}}}. \tag{9}$$

Although the Thurstonian discriminal process is typically applied using normal distributions, we extend this process to the models previously introduced, including the Bradley-Terry model (which

---

[1]This follows from $\text{var}(x_i - x_j) = \text{var}(x_i) + \text{var}(x_j) - 2\,\text{cov}(x_i, x_j)$.

uses the logistic distribution) and the Rao-Kupper and Davidson models (which incorporate ties). To our knowledge, these extensions have not been explored in the literature.

To generalize the Bradley-Terry model with the discriminal process, we assume $y_{ij}$ follows a logistic distribution, yielding

$$P(i \succ j \mid \{i,j\}) = \frac{1}{1 + e^{-z_{ij}}}, \quad \text{and} \quad P(i \prec j \mid \{i,j\}) = \frac{1}{1 + e^{-z_{ji}}}. \tag{10}$$

We can similarly extend the Rao-Kupper and Davidson models. The probabilities in (4) and (5) can be expressed in terms of the logit function $\log(\pi_i/\pi_j) = x_{ij}$, which represents the quantile of the logistic distribution. To incorporate the Thurstonian model, we replace $x_{ij}$ with $z_{ij}$. Thus, the Rao-Kupper model becomes

$$P(i \succ j \mid \{i,j\}) = \frac{1}{1 + e^{-(z_{ij} - \eta_{ij})}}, \tag{11a}$$

$$P(i \prec j \mid \{i,j\}) = \frac{1}{1 + e^{-(z_{ji} - \eta_{ji})}}, \tag{11b}$$

$$P(i \sim j \mid \{i,j\}) = \frac{e^{2\eta_{ij}} - 1}{\left(1 + e^{-(z_{ij} - \eta_{ij})}\right)\left(1 + e^{-(z_{ji} - \eta_{ji})}\right)}, \tag{11c}$$

where in the above we also used $\eta_{ij}$ to account for the generalized tie thresholds introduced in Section 2.3. Similarly, the Davidson model becomes

$$P(i \succ j \mid \{i,j\}) = \frac{1}{1 + e^{-z_{ij}} + e^{-(\frac{1}{2}z_{ij} - \eta_{ij})}}, \tag{12a}$$

$$P(i \prec j \mid \{i,j\}) = \frac{1}{1 + e^{-z_{ji}} + e^{-(\frac{1}{2}z_{ji} - \eta_{ji})}}, \tag{12b}$$

$$P(i \sim j \mid \{i,j\}) = \frac{1}{1 + e^{-(\frac{1}{2}z_{ij} + \eta_{ij})} + e^{-(\frac{1}{2}z_{ji} + \eta_{ji})}}. \tag{12c}$$

These models introduce an additional $m(m+1)/2$ covariance parameters $\sigma_{ij}$, which can lead to overparameterization. To address this, Thurstone (1927) proposed various constraints on the covariance matrix, while Takane (1989) suggested a factor model for covariance structure. We adopt the factor model employed by Böckenholt (1993); Maydeu-Olivares & Böckenholt (2005), where the covariance is constructed as

$$\boldsymbol{\Sigma} = \mathbf{D} + \boldsymbol{\Lambda}\boldsymbol{\Lambda}^{\mathsf{T}}, \tag{13}$$

where $\mathbf{D} = [d_{ij}]$ is a positive-definite diagonal matrix with $d_{ii} > 0$, and $\boldsymbol{\Lambda} = [\lambda_{ij}]$ is an $m \times k_{\mathrm{cov}}$ matrix of rank $k_{\mathrm{cov}} \leq m$, containing the factor parameters $\lambda_{ij}$. By selecting $k_{\mathrm{cov}}$, we can balance model fit with complexity.

The covariance matrix $\boldsymbol{\Sigma}$ is inherently non-unique, with multiple covariance matrices corresponding to the same variance matrix $\mathbf{S}$, forming an equivalence class. Additional details on the non-uniqueness, equivalence classes, and identifiability of covariance are provided in Appendix C.1.

## 2.5 Imposing Constraints to Resolve Symmetries

When estimating the parameters of the models (such as $\boldsymbol{\theta} = (\boldsymbol{x}, \mathbf{G})$ in the generalized tie models, or $\boldsymbol{\theta} = (\boldsymbol{\mu}, \mathbf{G}, \mathbf{D}, \boldsymbol{\Lambda})$ in their Thurstonian representation) through the maximum likelihood method described in (1), we encounter computational issues due to the non-uniqueness of the solution. This arises from the fact that the likelihood function remains invariant under certain transformations of the parameters, known as symmetries. These symmetries can result in poor optimization behavior, such as large or small parameter values, causing instability in the estimation process. To address these issues, we propose constraints on the log-likelihood function to eliminate the problematic symmetries. While one of these symmetries has been addressed in prior work, the other two have not, to the best of our knowledge.

The first symmetry arises from the model's invariance under the transformation $x_i \mapsto x_i + c$, where $c \in \mathbb{R}$ is a constant. This invariance implies that the values of $x_i$ (or $\mu_i$ in the Thurstonian representation) are not identifiable—only their differences are meaningful (see Appendix B for a detailed

discussion on parameter identifiability). Consequently, the parameters can shift uniformly without affecting the model's predictions. To address this symmetry, common approaches include fixing one of the score parameters, as demonstrated by Chiang et al. (2024), or imposing other constraints on the parameter values. For example, the original Bradley-Terry model (Bradley & Terry, 1952) enforces the constraint $\sum_{i=1}^{m} \pi_i = 1$. In our work, we adopt a similar approach by setting the mean of the scores: $\sum_{i=1}^{m} x_i = 0$, which corresponds to $\sum_{i=1}^{m} \mu_i = 0$ in the Thurstonian representation. This constraint effectively eliminates the shift invariance, ensuring stable optimization of the score parameters $\boldsymbol{x}$ (or $\boldsymbol{\mu}$).

The second symmetry, which has not been addressed in previous studies, involves scaling both the score and covariance parameters. Specifically, the Thurstonian models remain invariant under the transformation $(\mu_i, \sigma_{ij}) \mapsto (t\mu_i, t^2\sigma_{ij})$ for any positive constant $t$, because the ratio $z_{ij}$ in (9) remains unchanged. This symmetry can lead to parameters collapsing to very small values, causing numerical instability or underflow during optimization. To resolve this, we introduce the following constraint

$$\text{trace}(\tilde{\boldsymbol{\Sigma}}) = 1, \tag{14}$$

where $\tilde{\boldsymbol{\Sigma}} := \mathbf{P}\boldsymbol{\Sigma}\mathbf{P}$, and $\mathbf{P} := \mathbf{I} - \frac{1}{m}\mathbf{1}\mathbf{1}^{\intercal}$ is the centering operator with $\mathbf{I}$ as the identity matrix and $\mathbf{1} := (1, \ldots, 1)^{\intercal}$ is a column vector of ones. This constraint ensures proper scaling of the covariance parameters and avoids collapse during optimization. A detailed explanation of this constraint is provided in Appendix C.4.

The third symmetry pertains to the factor model parameters $\lambda_{ij}$. Using the factor model for covariance in (13), we can express the elements of the matrix $\mathbf{S}$ as

$$s_{ii} = 0, \quad \text{and} \quad s_{ij} = d_{ii} + d_{jj} + \|\boldsymbol{\lambda}_i - \boldsymbol{\lambda}_j\|_2^2, \quad i \neq j,$$

where $\boldsymbol{\lambda}_i := (\lambda_{i1}, \ldots, \lambda_{ik_{\text{cov}}})$ is the $i$-th row of the matrix $\boldsymbol{\Lambda}$, and $\|\cdot\|_2$ denotes the Euclidean norm. This expression reveals an invariance under translation $\lambda_{ij} \mapsto \lambda_{ij} + c$, which has not been addressed in the literature. To eliminate this translation symmetry, we impose the constraint

$$\|\boldsymbol{\Lambda}^{\intercal}\mathbf{1}\|_2^2 = 0. \tag{15}$$

This constraint fixes the column-wise mean of $\boldsymbol{\Lambda}$ to zero, preventing the factor parameters from arbitrarily shifting.

# 3 EMPIRICAL EVALUATION OF STATISTICAL MODELS

In this section, we evaluate the statistical models introduced earlier using the Chatbot Arena dataset. As of September 2024, the dataset comprises $m = 129$ competitors, with $|E| = 3455$ unique pairs. The total number of comparisons across all pairs is $\sum_{\{i,j\}\in E} n_{ij} = 1,374,996$, distributed as follows: 43.3% wins, 36.2% losses, and 20.4% ties.

We analyzed 30 configurations of the Bradley-Terry, Rao-Kupper, and Davidson models, detailed in Table D.1 in Appendix D.1. These configurations include both the original forms of the models and various generalizations introduced in this work, each assigned a unique ID corresponding to their rows in the table (e.g., Model 1, Model 2, etc.), which we reference throughout this and subsequent sections. For example, Model 1 corresponds to the Bradley-Terry model with ties treated as half a win and half a loss, following Chiang et al. (2024), while Model 4 represents the original Bradley-Terry model without ties. Similarly, Models 7 and 19 correspond to the original Rao-Kupper and Davidson models, respectively, with the remaining configurations representing our proposed generalizations.

Details of the experimental setup, along with results on model selection, goodness of fit, and generalization performance, are provided in Appendix D. Here, we focus on visual comparisons of pairwise win/loss and tie matrices to evaluate how well the models replicate observed outcomes across competitor pairs and to highlight their predictive capabilities.

## 3.1 EVALUATING THE PREDICTION OF WIN/LOSS AND TIE MATRICES

Figure 1 visualizes a subset of the win/loss matrix $\mathbf{W}$ and tie matrix $\mathbf{T}$ for the top 25 competitors predicted by four models. The first row of the figure shows win probabilities, while the second row

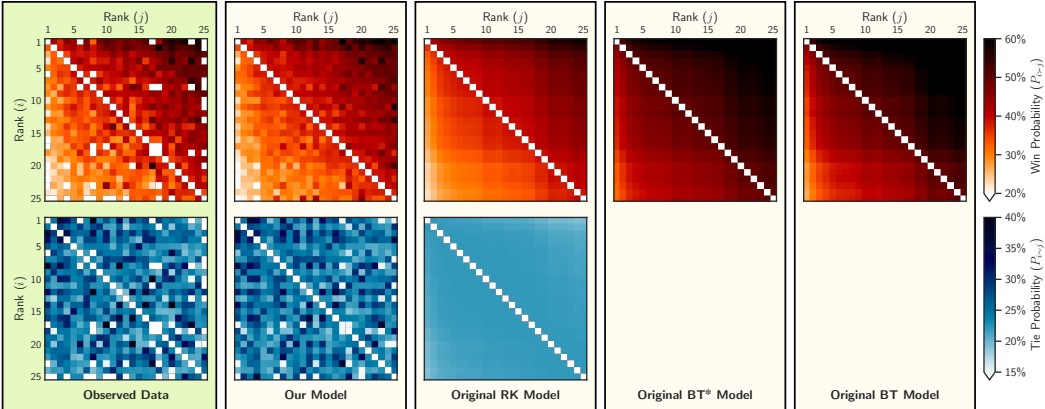

Figure 1: Comparison of pair-specific win (first row) and tie (second row) probabilities among 25 competitors between observed data (first column) and model predictions: our generalized Rao-Kupper with factored tie model (second column), original Rao-Kupper (RK) with ties (third column), Bradley-Terry with ties as half win/loss (fourth column, (Chiang et al., 2024)), and original Bradley-Terry without ties (fifth column). The ordinate and abscissa are shared across panels, shown only for the left-most and top-most panels. Each row shares the same color range and colorbar.

shows tie probabilities. The leftmost column contains the observed matrices, while the subsequent columns represent predictions from: the generalized Rao-Kupper model with factored ties (second column, corresponding to Model 18), the original Rao-Kupper (RK) model with a single tie parameter (third column, corresponding to Model 7), the Bradley-Terry model with ties treated as half win and half loss (Chiang et al., 2024) (fourth column, corresponding to Model 1), and the original Bradley-Terry model without ties (fifth column, corresponding to Model 4). Note that the Bradley-Terry models do not account for ties, so their tie probability matrices are absent. For consistency, the 25 competitors are ordered identically across all panels based on the ranking from Model 18.

The generalized Rao-Kupper model with factored ties (second column) exhibits fine-grained accuracy, closely matching the observed matrices for both win/loss and tie outcomes. In contrast, the original Rao-Kupper model (third column) predicts the win matrix reasonably well but shows reduced accuracy for ties. The Bradley-Terry models (fourth and fifth columns, with ties treated as half win/loss and without ties, respectively) differ significantly in their win/loss matrices, as they do not account for ties, resulting in noticeable discrepancies compared to the observed data.

To complement the pairwise probability comparisons presented here, Appendix D.5 evaluates the models using marginal probabilities of win, loss, and tie aggregated across all competitors.

## 4 RANKING AND COMPARISON OF LLM-BASED CHATBOTS

In this section, we demonstrate how the proposed models provide insights into the competitive performance of chatbots. Section 4.1 focuses on ranking analysis, while Section 4.2 explores relationships between chatbots, building on the covariance structures introduced in Section 2.4.

### 4.1 RANKING

In pairwise comparison methods, the score parameters $x$ (or $\mu$ in the Thurstonian representation) rank competitors, offering a straightforward interpretation of performance, with higher scores indicating stronger competitors. For example, Figure E.1 in Appendix E.1 shows the scores of the top 50 chatbots, ranked by Model 18 in Table D.1 (generalized Rao-Kupper model with $k_{\text{cov}} = 3$ and $k_{\text{tie}} = 20$).

A key question is how different statistical models influence these rankings. To address this, we analyzed rankings produced by 12 selected models from Table D.1. Figure E.2 in Appendix E.2 visualizes the bump chart of these rankings across models of varying complexity. Each column

represents a model, ordered from simpler models on the right to more complex ones on the left, and each row tracks a chatbot's rank across models.

Notably, top-ranked chatbots exhibit stable rankings across all models, indicating reliability regardless of model complexity. In contrast, lower-ranked chatbots display greater variation, reflecting their sensitivity to model selection.

To systematically analyze these differences, we computed the Kendall rank correlation ($\tau$) between all 30 models in Table D.1, as shown in Figure E.3 in Appendix E.3. Kendall's $\tau$ measures the agreement between two rankings by evaluating the relative ordering of pairs, making it particularly well-suited for ordinal data. This analysis revealed that rankings across models are highly correlated, but notable differences emerge based on model parameterizations.

To better interpret these differences, we applied hierarchical clustering to the Kendall correlation matrix, as detailed in Appendix E.4. The resulting ordering of models, displayed in Figure E.3, is determined directly by the optimal ordering suggested by the clustering algorithm, reflecting the structure it uncovered.

Interestingly, the inclusion or exclusion of the Thurstonian covariance factor emerges as the primary driver of these clusters: models without covariance (indicated by ✗) form a distinct cluster, separate from those with covariance. Furthermore, within the covariance group, models are divided into two subgroups based on the complexity of their covariance structures: $k_{\mathrm{cov}} = 0$ and $k_{\mathrm{cov}} = 3$. This highlights the strong influence of covariance parameters on ranking similarities.

While covariance prominently drives ranking—more so than other model features such as tie parameters—earlier results (see Section 3 and Appendix D) showed that generalized tie modeling significantly enhances model fit and predictive accuracy. These findings highlight complementary strengths: tie modeling improves fit, while covariance modeling shapes ranking patterns and distinguishes competitors based on shared performance characteristics.

## 4.2 EXPLORING COMPETITOR CORRELATIONS

Incorporating covariance into Thurstonian models allows for exploring relationships between competitors beyond simple rankings. While rankings provide ordinal insights, covariance structures quantify uncertainties and correlations, uncovering patterns that rankings alone cannot capture.

As detailed in Appendix C.1, the covariance matrix $\boldsymbol{\Sigma}$ is not unique, but its associated matrix $\mathbf{S} = [s_{ij}]$, representing the variance of score differences, is unique and interpretable. This matrix serves as a dissimilarity measure, capturing the relative uncertainty in performance comparisons. By normalizing score differences with $\sqrt{s_{ij}}$, we construct the matrix $\mathbf{Z} = [z_{ij}]$ as given in (9), which forms the basis for analyzing relationships among competitors.

To visualize these relationships, we apply kernel PCA to $\mathbf{Z}$ and project the data into three dimensions, as shown in Figure 2. Each point represents a competitor, with distances reflecting their dissimilarities. We use a squared exponential kernel, $\rho_{ij} = \exp(-\gamma z_{ij}^2)$, with $\gamma = 10^{-4}$, to improve interpretability in feature space. This spatial representation highlights groupings of competitors with similar performance patterns.

Complementing the kernel PCA visualization in Figure 2, Appendix C.2 employs multidimensional scaling (MDS) as an alternative approach for interpreting the dissimilarity matrix $\mathbf{Z}$. MDS directly translates pairwise dissimilarities into spatial distances, preserving relational structures in a reduced-dimensional space. Both KPCA and MDS uncover consistent spatial grouping patterns that align closely with rankings.

To systematically capture these spatial patterns, we applied hierarchical clustering to $\mathbf{Z}$, as detailed in Appendix C.3. The clustering analysis organizes competitors into performance-based tiers. For example, top performers such as ChatGPT-4o Latest and Gemini 1.5 Experimental form distinct clusters, reflecting their dominance, while mid- and lower-ranked competitors stratify into meaningful subgroups. Notably, the optimal ordering in hierarchical clustering effectively retrieves the ranking order, even without direct access to individual scores. These results emphasize the value of covariance in analyzing relational patterns.

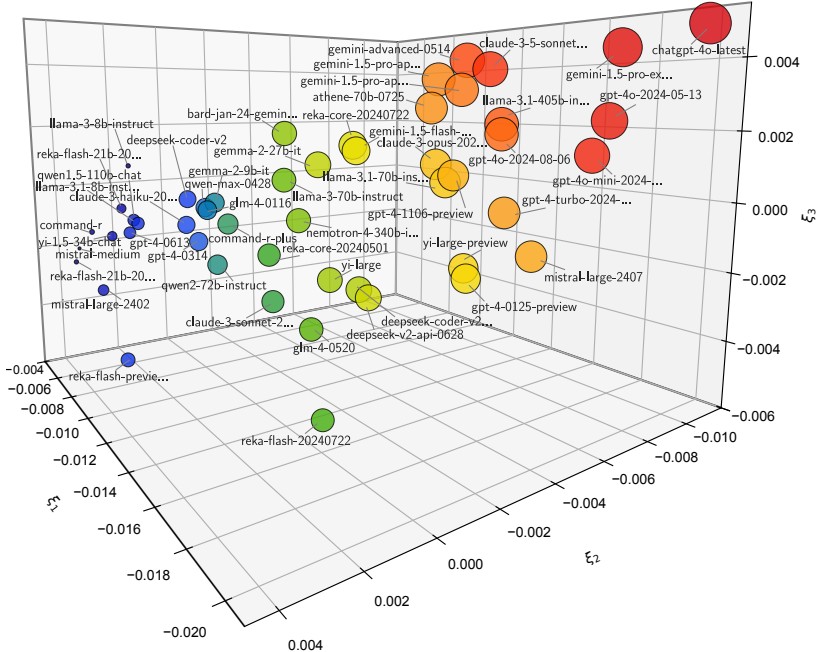

Figure 2: Kernel PCA projection of the distance matrix $\mathbf{Z}$ onto three dimensions, focusing on the top 40 competitors ranked by Model 18 of Table D.1. Circle size and color are proportional to the competitors' scores.

## 5   CONCLUSION

Evaluating large language models (LLMs) through pairwise comparisons is essential for assessing their capabilities and limitations in practical applications. This paper presents a statistical framework that extends traditional methods by incorporating tie modeling, covariance structures, and constraints to address optimization challenges, enhancing interpretability, stability, and accuracy in LLM evaluations.

Our generalized tie model addresses long-standing limitations of Rao-Kupper and Davidson methods, reducing prediction errors by two orders of magnitude and improving fit across win, loss, and tie outcomes. By introducing covariance structures, our framework uncovers latent relationships among competitors, enabling clustering into performance tiers and enriching interpretability beyond rankings. Additionally, we resolve optimization challenges through novel constraints, ensuring stable and unique parameter estimation.

A key insight from our analysis is that covariance structures not only enrich interpretability but also shape ranking consistency, underscoring their critical role in capturing nuanced relationships in pairwise comparisons. This finding highlights the broader utility of covariance-based approaches in complex evaluation tasks.

To validate our framework, we conducted rigorous empirical evaluations, demonstrating significant improvements in model fit and interpretability compared to existing methods. To support reproducibility and broader adoption, we provide `leaderbot`, an open-source Python package implementing our statistical framework with tools for data processing, model fitting, and visualization.

Our work not only advances LLM evaluation but also provides a robust foundation for analyzing pairwise comparison data in diverse domains. By bridging theoretical rigor with practical applicability, this framework opens new avenues for ranking, inference, and interpretability in complex comparison tasks.

ACKNOWLEDGMENTS

The authors thank Anastasios Nikolas Angelopoulos and Tianle Li for their thoughtful feedback. MWM acknowledges support from the DOE, IARPA, NSF, and ONR.

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

# Appendices

CONTENTS

## APPENDIX A    BROADER IMPLICATIONS OF PAIRED COMPARISON METHODS

Paired comparison frameworks are foundational in many fields and have been widely applied in classical and modern domains. In sports analytics, they are used to predict match outcomes and assess

player performance in games such as chess (Elo, 1978), tennis (Glickman, 1995), and soccer. Marketing applications include optimizing product offerings, advertisement placements, and analyzing consumer preferences. In psychometrics and behavioral studies, paired comparisons assess perception and attitudes in response to visual or auditory stimuli. Similarly, in election studies and political science, they are employed to rank candidates, analyze voting behavior, test referendum arguments (Loewen et al., 2012), and measure perceived political ideologies (Hopkins & Noel, 2022). Clinical research also uses paired comparisons to evaluate treatments and interventions in clinical trials and epidemiological studies. These diverse applications illustrate the versatility of paired comparison frameworks in extracting meaningful inferences from comparative data.

Recent advancements have extended paired comparison methods to machine learning, where they play a pivotal role in preference modeling and optimization. For instance, Reinforcement Learning with Human Feedback (RLHF) uses paired comparisons to fine-tune large language models by ranking outputs based on human preferences, often employing the Bradley-Terry model for preference quantification (Rafailov et al., 2024; Karthik et al., 2024). Direct Preference Optimization (DPO) further refines this approach by aligning model outputs directly with human preferences without relying on scalar reward models (Wu et al., 2023). Additionally, methodologies like Pairwise Proximal Policy Optimization (P3O) leverage relative feedback to enhance LLM alignment (Wu et al., 2024). Innovations such as the integration of Rao-Kupper models have enabled paired comparison frameworks to incorporate ties, capturing ambiguous or neutral preferences in RLHF settings (Liu et al., 2024). These developments highlight the growing influence of paired comparison methods in machine learning and underscore the potential of our generalizations to enhance these frameworks further.

## APPENDIX B   UNIDENTIFIABILITY OF PARAMETERS IN PAIRED COMPARISON MODELS

In this section, we discuss the challenge of estimating the uncertainties of scores $x_i$ in paired comparison models. Previous works, such as Chiang et al. (2024), have introduced methods for computing confidence intervals for scores using empirical approaches like bootstrapping. While these methods can be valuable in practice, we demonstrate that this problem is intrinsically ill-posed due to the unidentifiability of the score parameters. Specifically, the likelihood function depends only on score differences $x_i - x_j$, rendering individual score estimates invariant under shift transformation. This invariance introduces non-uniqueness in the quantification of confidence intervals.

This issue arises not from specific modeling choices but from a structural characteristic of models based on linear stochastic transitivity, where probabilities take the form $F(x_i - x_j)$ (see Section 2.2). To analyze this limitation rigorously, we examine the Fisher Information Matrix (FIM) of the likelihood function. In Appendix B.1, we provide a background on the FIM and its role in parameter identifiability. In Appendix B.2, we show that the score parameters are unidentifiable due to the singularity of the FIM. Finally, in Appendix B.3, we propose reparameterizations that result in identifiable quantities.

### B.1   FISHER INFORMATION AND IDENTIFIABILITY: BACKGROUND

The Fisher Information Matrix (FIM) quantifies the amount of information that the likelihood function carries about the parameters of interest. It can be derived from the gradient of the log-likelihood function, known as the informant vector,[2] or equivalently, from the negative Hessian matrix of the log-likelihood under mild regularity conditions:

$$\mathbf{F}(\boldsymbol{\theta}) := \mathbb{E}\left[\nabla_{\boldsymbol{\theta}}\ell(\boldsymbol{\theta}) \otimes \nabla_{\boldsymbol{\theta}}\ell(\boldsymbol{\theta}) \,|\, \boldsymbol{\theta}\right] = -\mathbb{E}\left[\nabla_{\boldsymbol{\theta}}\nabla_{\boldsymbol{\theta}}^{\mathsf{T}}\ell(\boldsymbol{\theta}) \,|\, \boldsymbol{\theta}\right]. \tag{B.1}$$

The FIM measures the curvature of the likelihood function around the estimated parameters, reflecting the precision of the parameter estimates. A sharper likelihood function implies higher confidence in the parameter estimates. Formally, the Cramér-Rao bound establishes a theoretical lower bound for the covariance of the parameter estimates (see e.g., Söderström & Stoica (1989)):

$$\mathrm{cov}(\boldsymbol{\theta}) \geq \mathbf{F}^{-1}(\boldsymbol{\theta}). \tag{B.2}$$

---

[2]Commonly referred to as the *score* but this term is avoided here to prevent confusion with the score parameters $x_i$.

This lower bound is often used to derive estimates of parameter uncertainty. For example, assuming the approximation $\mathrm{var}(\theta_i) \approx [\mathbf{F}^{-1}]_{ii}$, the confidence interval for $\theta_i$ can be estimated as

$$\Delta\theta_i = t_{\alpha,n-m}\sqrt{\mathrm{var}(\theta_i)}, \tag{B.3}$$

where $t_{\alpha,n-m}$ is the critical value from the Student's $t$-distribution for a confidence level $\alpha \in [0,1]$ with $n-m$ degrees of freedom, $n$ being the number of data points and $m$ the number of parameters. This approach yields a conservative estimate of the variance of $\theta_i$, reflecting the Cramér-Rao bound. Alternative methods, such as bootstrapping, may provide more practical confidence intervals in certain cases.

When the FIM is ill-conditioned or singular, however, the parameter estimation problem becomes ill-posed. In such cases, the uncertainty bounds $\Delta\theta_i$ become unbounded or undefined. This occurs when the likelihood function exhibits invariance under certain transformations of the parameters, leading to parameter redundancy. We formally define parameter identifiability and its connection to the FIM below.

**Definition B.1** ((Rothenberg, 1971, Definitions 1, 2, and 3))**.** Two parameter vectors $\boldsymbol{\theta}$ and $\boldsymbol{\theta}'$ are said to be *observationally equivalent* if $\ell(\boldsymbol{\theta}) = \ell(\boldsymbol{\theta}')$. A parameter vector $\boldsymbol{\theta}$ is *locally identifiable* if there exists an open neighborhood around $\boldsymbol{\theta}$ containing no other $\boldsymbol{\theta}'$ that is observationally equivalent to $\boldsymbol{\theta}$. If $\boldsymbol{\theta}$ is not observationally equivalent to any other parameter vector in the entire domain of the likelihood function, it is said to be *globally identifiable*.

**Theorem B.1** ((Rothenberg, 1971, Theorem 1))**.** *Suppose the Fisher Information Matrix $\mathbf{F}$ is a continuous function of $\boldsymbol{\theta}$. Let $\boldsymbol{\theta}^*$ be a regular point of $\mathbf{F}$, meaning $\mathbf{F}(\boldsymbol{\theta})$ has constant rank in a neighborhood of $\boldsymbol{\theta}^*$. Then, $\boldsymbol{\theta}^*$ is locally identifiable if and only if $\mathbf{F}(\boldsymbol{\theta}^*)$ is non-singular.*

The above theorem establishes that the FIM plays a central role in determining parameter identifiability (see also (Seber & Wild, 2005, Sections 3.4 and 8.4)).

### B.2 UNIDENTIFIABILITY OF SCORE PARAMETERS

We now focus on the identifiability of the score parameters, $\boldsymbol{x}$, in paired comparison models. For simplicity, we limit the analysis to $\boldsymbol{x}$, though the results extend naturally to other parameters. We prove that the FIM for $\boldsymbol{x}$ is singular for likelihood functions satisfying the shift invariance property. While the invariance property trivially implies unidentifiability by definition, analyzing the FIM reveals deeper insights into the parameter space. Specifically, it identifies the null space causing unidentifiability and highlights subspaces suitable for well-defined reparametrizations, as explored in the next subsection.

**Proposition B.1.** *Let the log-likelihood function $\ell \in C^2(\mathbb{R}^m, \mathbb{R})$ satisfy the shift invariance property*

$$\ell(\boldsymbol{x} + c\mathbf{1}) = \ell(\boldsymbol{x}), \quad c \in \mathbb{R}. \tag{B.4}$$

*Then, the corresponding Fisher Information Matrix $\mathbf{F}(\boldsymbol{x})$ is singular where $\mathrm{rank}(\mathbf{F}(\boldsymbol{x})) \le m-1$, with $\mathbf{1}$ (the vector of ones) in its null space.*

**Proof.** From (B.4) we have

$$\frac{\partial\ell(\boldsymbol{x} + c\mathbf{1})}{\partial c} = \sum_{j=1}^m \frac{\partial\ell(\boldsymbol{x})}{\partial x_j} = 0. \tag{B.5}$$

On the other hand, summing over all columns of the Hessian, $\mathbf{H} := \nabla_{\boldsymbol{x}}\nabla_{\boldsymbol{x}}^{\mathsf{T}}\ell(\boldsymbol{x})$, and using (B.5) yields

$$\sum_{j=1}^m H_{ij} = \sum_{j=1}^m \frac{\partial^2\ell(\boldsymbol{x})}{\partial x_i \partial x_j} = \frac{\partial}{\partial x_i}\left(\sum_{j=1}^m \frac{\partial\ell(\boldsymbol{x})}{\partial x_j}\right) = 0, \qquad \forall\, i = 1, \dots, m. \tag{B.6}$$

Hence, $\mathbf{H}$ has a zero row sum, implying $\mathbf{H1} = \mathbf{0}$. Therefore, $\mathbf{1}$ lies in the null space of $\mathbf{H}$, and by extension, $\mathbf{F}(\boldsymbol{x})$ is singular with a rank of at most $m-1$. $\qquad\square$

The singularity of $\mathbf{F}(\boldsymbol{x})$ confirms the unidentifiability of the score parameters $\boldsymbol{x}$, rendering the quantification of their uncertainties fundamentally ill-posed. As a result, the lower bounds from the Cramér-Rao inequality in (B.2) become unbounded, making the confidence interval such as in (B.3) undefined. In the next section, we analyze the structure of $\mathbf{F}(\boldsymbol{x})$ to identify subspaces where meaningful parameter estimation is possible.

### B.3 IDENTIFIABLE PARAMETRIZATION

We now address which quantities are identifiable through the FIM. Specifically, any reparameterization within the range of the FIM is identifiable. Let $\mathcal{N}_{\boldsymbol{\theta}}$ denote the null space of the FIM and $\mathcal{N}_{\boldsymbol{\theta}}^{\perp}$ its orthogonal complement. Suppose $\boldsymbol{\theta} = \boldsymbol{\theta}^*$ is a local minima of the likelihood function. The FIM, when restricted to $\mathcal{N}_{\boldsymbol{\theta}^*}^{\perp}$, is positive definite, and any parameterization within this subspace is identifiable.

In the case of pairwise comparison with the optimal solution $\boldsymbol{x} = \boldsymbol{x}^*$, assuming $\mathrm{rank}(\mathbf{F}(\boldsymbol{x}^*)) = m - 1$, we have $\mathcal{N}_{\boldsymbol{x}^*} \coloneqq \mathrm{span}(\mathbf{1})$. The projection operator onto $\mathcal{N}_{\boldsymbol{x}^*}^{\perp}$ is given by

$$\mathbf{P} = \mathbf{I} - \frac{1}{m}\mathbf{1}\mathbf{1}^{\intercal}, \tag{B.7}$$

which is the centering matrix that converts $\boldsymbol{x}^*$ to the mean-zero vector $\tilde{\boldsymbol{x}}^* \coloneqq \mathbf{P}\boldsymbol{x}^* \in \mathcal{N}_{\boldsymbol{x}^*}^{\perp}$. A representation of this reparameterization is provided by the $(m-1) \times m$ *forward differencing matrix* $\mathbf{A} : \mathbb{R}^m \to \mathcal{N}_{\boldsymbol{x}^*}^{\perp}$, with entries $A_{i,i} = 1$, $A_{i,i+1} = -1$, and zero otherwise. Using $\mathbf{A}$, $\boldsymbol{x}^*$ can be transformed into $\boldsymbol{y}^* \coloneqq \mathbf{A}\boldsymbol{x}^*$, where each component $y_i^* = x_i^* - x_{i+1}^*$ represents the difference between adjacent elements of $\boldsymbol{x}^*$. This reparameterization lies entirely in $\mathcal{N}_{\boldsymbol{x}^*}^{\perp}$, making $\boldsymbol{y}^*$ identifiable and enabling meaningful quantification of its uncertainty.

Thus, in paired comparison models we consider, only differences in scores provide meaningful inference. In the next section, we explore this in the context of Thurstonian covariance parameters.

## APPENDIX C COVARIANCE MODEL

In Section 2.4 we expand the inclusion of covariance via Thurstonian model. We recall that, in Thurstonian model, the score parameters are assumed to be stochastic with $x_i = \mu_i + \epsilon_i$ where $\epsilon_i$ is the stochastic component with the covariance $\boldsymbol{\Sigma} = [\sigma_{ij}]$ where $\sigma_{ij} \coloneqq \mathrm{cov}(\epsilon_i, \epsilon_j)$. Furthermore, we defined the matrix $\mathbf{S} = [s_{ij}]$ where $s_{ij} = \sigma_{ii} + \sigma_{jj} - 2\sigma_{ij}$ representing the variance of $x_i - x_j$.

In this section, we provide a detailed analysis of the covariance matrix $\boldsymbol{\Sigma}$ and its associated matrix $\mathbf{S}$. In particular, in Appendix C.1, we explore the identifiability, particularly how $\boldsymbol{\Sigma}$ is inherently non-unique while $\mathbf{S}$ remains unique and identifiable. In Appendix C.2 we present how to interpret and visualize these matrices. In Appendix C.3, we examine hierarchical clustering of competitors based on the dissimilarity matrix, uncovering performance tiers and relationships. Finally, in Appendix C.4 we discuss constraints that allow stable identification of covariance during optimization of likelihood.

### C.1 NON-UNIQUENESS AND EQUIVALENCE CLASS OF COVARIANCE

We begin by noting that the likelihood function in the Thurstonian models we presented depends on the function of $z_{ij}$ defined in (9), which itself depends on $s_{ij}$. That is, $\mathbf{S}$ is an observable quantity, while $\boldsymbol{\Sigma}$ is a latent variable. Below, we formalize the relationship between these two matrices and the equivalence class of covariance matrices that share the same $\mathbf{S}$.

Let $\mathbb{S}^m$ denote the space of symmetric $m \times m$ matrices and $\mathbb{S}_{\circ}^m$ be the the space of hollow symmetric matrices where all diagonal elements are zero. Define the map $\mathcal{S} : \mathbb{S}^m \to \mathbb{S}_{\circ}^m$ that associates a covariance matrix $\boldsymbol{\Sigma}$ with the matrix $\mathbf{S}$, given by

$$\mathbf{S} = \mathcal{S}(\boldsymbol{\Sigma}) = \mathrm{diag}(\boldsymbol{\Sigma})\mathbf{1}^{\intercal} + \mathbf{1}\,\mathrm{diag}(\boldsymbol{\Sigma})^{\intercal} - 2\boldsymbol{\Sigma}, \tag{C.1}$$

where $\mathrm{diag}(\boldsymbol{\Sigma})$ is a vector containing the diagonal elements of $\boldsymbol{\Sigma}$. This relation corresponds to $s_{ij} = \sigma_{ii} + \sigma_{jj} - 2\sigma_{ij}$ in matrix form.

As we will show momentarily, the map $\mathcal{S}$ is non-injective, as for each $\mathbf{S}$, there exist non-unique covariance matrices $\boldsymbol{\Sigma}$ differing by elements in the kernel of $\mathcal{S}$, all of which map to the same $\mathbf{S}$. Consequently, the preimage of $\mathcal{S}$ defines the equivalence class of covariance matrices producing the same $\mathbf{S}$, given by

$$[\boldsymbol{\Sigma}] = \mathcal{S}^{-1}(\mathbf{S}) = \left\{ \boldsymbol{\Sigma}' \in \mathbb{S}^m \mid \mathcal{S}(\boldsymbol{\Sigma}') = \mathbf{S} \right\}. \tag{C.2}$$

This equivalence class partitions $\mathbb{S}^m$ modulo the kernel of $\mathcal{S}$, denoted as $\mathbb{S}^m / \ker(\mathcal{S})$. We now formalize this structure.

**Proposition C.1** (Equivalence Class of Covariance). *The map $\mathcal{S} : \mathbb{S}^m \to \mathbb{S}_\circ^m$, defined in (C.1), is a surjective, non-injective linear transformation. Its kernel is given by*

$$\ker(\mathcal{S}) = \{ \boldsymbol{v} \mathbf{1}^\mathsf{T} + \mathbf{1} \boldsymbol{v}^\mathsf{T} \mid \boldsymbol{v} \in \mathbb{R}^m \}. \tag{C.3}$$

*Consequently, the quotient space $\mathbb{S}^m / \ker(\mathcal{S})$ represents the space of equivalence classes of covariance matrices of the form*

$$[\boldsymbol{\Sigma}] = \{ \boldsymbol{\Sigma} + \boldsymbol{v} \mathbf{1}^\mathsf{T} + \mathbf{1} \boldsymbol{v}^\mathsf{T} \mid \boldsymbol{v} \in \mathbb{R}^m \}, \tag{C.4}$$

*where all elements of $[\boldsymbol{\Sigma}]$ map to the same matrix $\mathbf{S}$ under $\mathcal{S}$.*

**Proof.** To determine $\ker(\mathcal{S})$, consider $\boldsymbol{\Sigma}' \in \mathbb{S}^m$ such that $\mathcal{S}(\boldsymbol{\Sigma}') = \mathbf{0}$. From (C.1), it follows that

$$\operatorname{diag}(\boldsymbol{\Sigma}') \mathbf{1}^\mathsf{T} + \mathbf{1} \operatorname{diag}(\boldsymbol{\Sigma}')^\mathsf{T} - 2\boldsymbol{\Sigma}' = \mathbf{0}.$$

Rearranging, we find

$$\boldsymbol{\Sigma}' = \frac{1}{2} \left( \operatorname{diag}(\boldsymbol{\Sigma}') \mathbf{1}^\mathsf{T} + \mathbf{1} \operatorname{diag}(\boldsymbol{\Sigma}')^\mathsf{T} \right),$$

implying that any $\boldsymbol{\Sigma}' \in \ker(\mathcal{S})$ must be of the form given in (C.3). The equivalence class $[\boldsymbol{\Sigma}] = \mathcal{S}^{-1}(\mathbf{S})$ is obtained by adding elements of $\ker(\mathcal{S})$ to a representative $\boldsymbol{\Sigma}$, yielding (C.4).

To show $\mathcal{S}$ is surjective, observe that for any $\mathbf{S} \in \mathbb{S}_\circ^m$, the matrix $-\frac{1}{2} \mathbf{S} \in \mathbb{S}^m$ satisfies $\mathcal{S}(-\frac{1}{2} \mathbf{S}) = \mathbf{S}$. Hence, every $\mathbf{S} \in \mathbb{S}_\circ^m$ has at least one preimage, proving surjectivity. $\qquad\square$

As demonstrated in Proposition C.1, the covariance matrix $\boldsymbol{\Sigma}$ is not unique, as adding any symmetric rank-one matrix to it leaves the likelihood invariant. Consequently, $\boldsymbol{\Sigma}$ is not identifiable by Definition B.1.

## C.2 Interpretation and Visualization of Covariance

While the covariance matrix $\boldsymbol{\Sigma}$ is not unique, the matrix $\mathbf{S}$ is unique and identifiable. This makes $\mathbf{S}$ the preferred object for interpreting relationships between competitors. Unlike $\boldsymbol{\Sigma}$, which represents similarity, $\mathbf{S}$ plays the role of a *dissimilarity matrix*, as commonly explored in paired comparison literature (Maydeu-Olivares & Böckenholt, 2005; Böckenholt, 2006). In fact, under suitable conditions, $\mathbf{S}$ can be interpreted as a distance matrix.

For $\mathbf{S}$ to qualify as a distance matrix, it must be non-negative. This holds if and only if the doubly-centered covariance matrix, defined as

$$\tilde{\boldsymbol{\Sigma}} = \mathbf{P} \boldsymbol{\Sigma} \mathbf{P}, \tag{C.5}$$

is positive semi-definite. Here, $\mathbf{P}$ is the centering matrix from (B.7). The matrix $\tilde{\boldsymbol{\Sigma}}$ represents the covariance of mean-centered scores, $\tilde{\boldsymbol{x}} = \mathbf{P} \boldsymbol{x}$. Importantly, $\tilde{\boldsymbol{\Sigma}} \in [\boldsymbol{\Sigma}]$ and is unique within the equivalence class $[\boldsymbol{\Sigma}]$. Specifically, for any $\boldsymbol{\Sigma}', \boldsymbol{\Sigma}'' \in [\boldsymbol{\Sigma}]$, it holds that $\mathbf{P} \boldsymbol{\Sigma}' \mathbf{P} = \mathbf{P} \boldsymbol{\Sigma}'' \mathbf{P} = \tilde{\boldsymbol{\Sigma}}$. This ensures that $\tilde{\boldsymbol{\Sigma}}$ is well-defined and serves as a canonical representation of the covariance structure.

A matrix $\mathbf{S}$ is called a Euclidean distance matrix if there exist spatial points $\boldsymbol{p}_1, \ldots, \boldsymbol{p}_m$ such that $s_{ij} = \| \boldsymbol{p}_i - \boldsymbol{p}_j \|^2$. Mardia et al. (1979, Theorem 14.2.1) guarantees that $\mathbf{S}$ is a Euclidean distance matrix if and only if $\tilde{\boldsymbol{\Sigma}}$ is positive semi-definite. In our setting, this condition is always satisfied, as we enforce the factor covariance model in (13) as

$$\boldsymbol{\Sigma} = \mathbf{D} + \boldsymbol{\Lambda} \boldsymbol{\Lambda}^\mathsf{T}, \tag{C.6}$$

where $\mathbf{D}$ is a positive diagonal matrix ($\mathbf{D} > \mathbf{0}$), ensuring that $\boldsymbol{\Sigma}$ is positive definite. Consequently, $\tilde{\boldsymbol{\Sigma}}$ is positive semi-definite, making $\mathbf{S}$ a Euclidean distance matrix.

This property enables meaningful visualization of $\mathbf{S}$ using multi-dimensional scaling (MDS). MDS (see e.g., (Mardia et al., 1979, Chapter 14) or (Seber & Wild, 2005, Section 5.5)) constructs a set of points in two- or three-dimensional space such that their pairwise distances approximate the distances in $\mathbf{S}$. This approach is particularly suitable for visualizing $\mathbf{S}$ or similar distance matrices derived from covariance models.

Here, we use the matrix $\mathbf{Z}$ as the distance matrix, as defined in (9), where $z_{ij} = (x_i - x_j)/\sqrt{s_{ij}}$, capturing both score differences and dissimilarities derived from covariance. The dissimilarity represented by $\mathbf{Z}$ is visualized in Figure C.1 using MDS, showing the first two principal coordinates in a

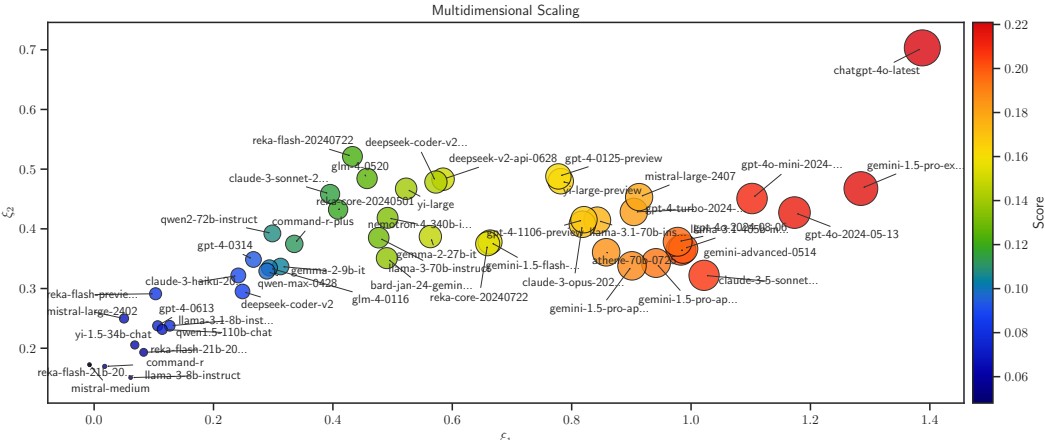

Figure C.1: MDS projection of the distance matrix $\mathbf{Z}$ onto two dimensions, focusing on the top 50 competitors ranked by Model 18 of Table D.1. In the scatter plot, circle size and color are proportional to the competitors' scores.

two-dimensional space. Interestingly, the spatial arrangement of points in the plot not only reflects the pairwise dissimilarities but also aligns well with the overall ranking of competitors, effectively capturing their relative scores. This highlights the ability of MDS to extract meaningful patterns from the dissimilarity matrix alone, without access to the values of the scores.

A companion approach to MDS is kernel PCA (Schölkopf et al., 1999; Williams, 2000), applied earlier in Section 4.2 and visualized in Figure 2. Both techniques aim to uncover relationships in lower-dimensional spaces, with kernel PCA operating on points in a feature space and MDS working directly with pairwise distances. These methods are dual representations: PCA identifies principal components of the data, while MDS identifies principal coordinates of a distance matrix (Mardia et al., 1979, Section 14.3). Notably, both the kernel PCA in Figure 2 (projected into three dimensions) and the MDS in Figure C.1 (projected into two dimensions) rely on the same dissimilarity matrix $\mathbf{Z}$, offering consistent visual representations of the relationships between competitors.

## C.3 HIERARCHICAL CLUSTERING OF COMPETITOR PERFORMANCE

To complement the analysis of dissimilarity using PCA and MDS, we applied hierarchical agglomerative clustering (Hastie et al., 2009, Section 14.3.12) with optimal leaf ordering (Bar-Joseph et al., 2001) to the dissimilarity matrix $\mathbf{Z}$. We recall that the matrix $\mathbf{Z}$ incorporates both score differences and dissimilarities derived from Thurstonian covariance in our generalized models. This integration of covariance within the model enhances the interpretability of the clustering results by capturing both the performance levels of competitors and their correlation structure.

The clustering analysis focuses on the top 100 competitors ranked using Model 18 from Table D.1. The resulting dendrogram, shown in Figure C.2, reveals a hierarchical structure that organizes competitors into distinct tiers. Interestingly, despite the clustering algorithm having access only to the dissimilarity matrix $\mathbf{Z}$, which encodes score differences (but not the values of the scores) and covariance-derived uncertainties, the clustering order closely aligns with the competitors' rankings. Each cluster roughly corresponds to a contiguous range of ranks, albeit with minor variations, demonstrating the ability of the clustering method to infer meaningful performance groupings solely from relative differences.

This hierarchical structure provides additional insights into the relationships between competitors, suggesting a natural stratification into performance-based tiers. Tier I includes the two leading models, ChatGPT-4o Latest (as of September 2024) and Gemini 1.5 Pro Experimental, reflecting their dominance. The remaining 98 models form Tier II, which is further divided into two subgroups: $II_A$ (green theme) and $II_B$ (red theme). These subgroups are further refined into smaller clusters:

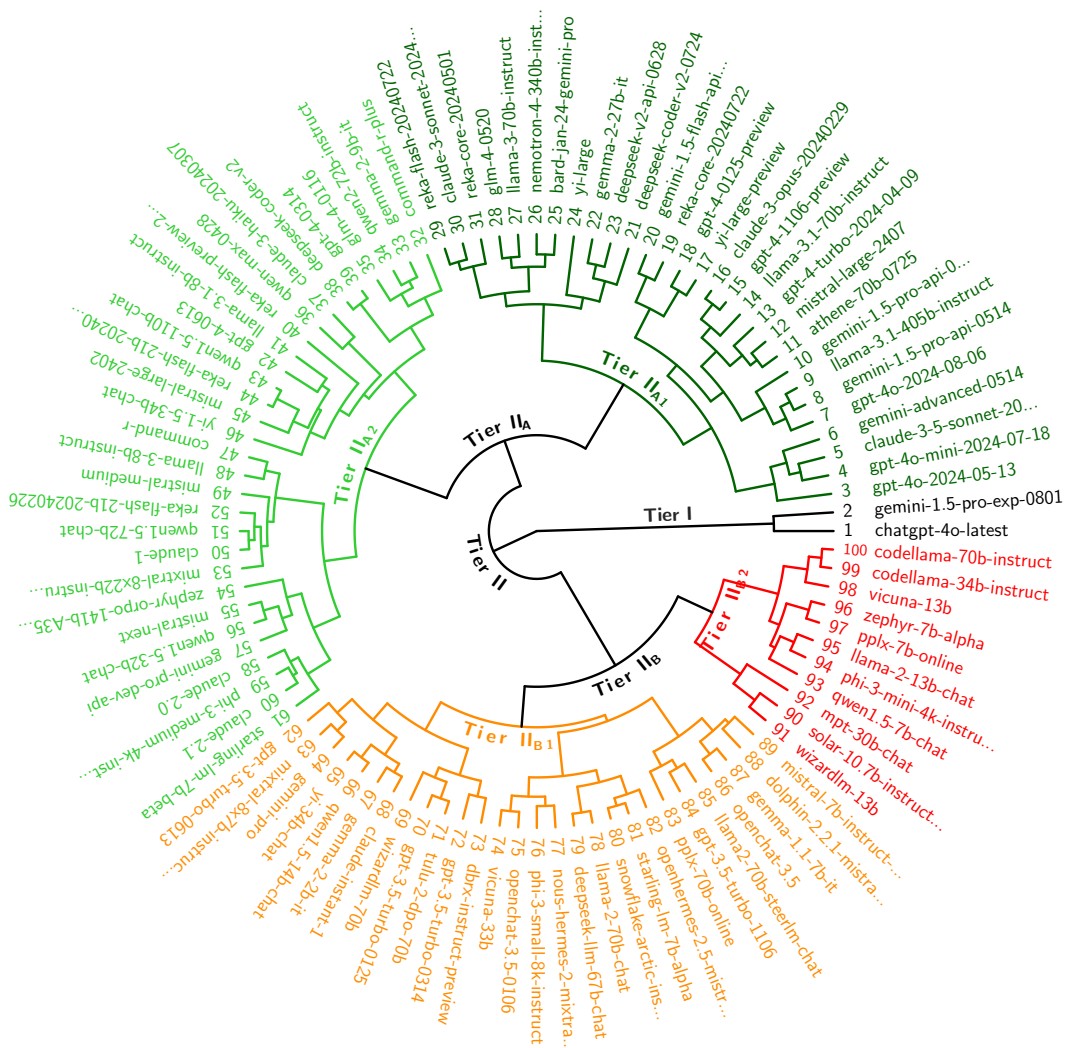

Figure C.2: Hierarchical clustering of the top 100 competitors based on the distance matrix **Z** derived from Model 18 of Table D.1. The clustering reveals performance tiers, with Tier I consisting of the top two competitors (ChatGPT-4 Latest and Gemini 1.5 Experimental) and Tier II further subdivided into groups representing decreasing performance levels. This structure highlights the relationships and relative strengths among competitors.

$II_A$ is split into $II_{A_1}$ (dark green) and $II_{A_2}$ (light green), while $II_B$ is split into $II_{B_1}$ (light red) and $II_{B_2}$ (dark red).

The meaningfulness of these groupings arises from the use of **Z**, which integrates performance scores with covariance-based dissimilarities—a capability enabled by incorporating Thurstonian models in our framework. This analysis complements rankings by uncovering hierarchical structures and relational patterns among competitors.

While our analysis focuses on hierarchical clustering derived from the statistical properties of the dissimilarity matrix, alternative clustering approaches, such as those leveraging semantic relationships (e.g., embeddings or linguistic features), could provide complementary insights into model relationships. Future research could explore these directions, integrating semantic and statistical perspectives to uncover deeper insights into competitor relationships.

### C.4 Constraint on Thurstonian Covariance Model

In Section 2.5 we presented constraints to resolve the symmetry of the likelihood functions with respect to transformations of the parameters. Here, we provide further detail on the second symmetry presented therein. Specifically, we recall that the models are invariant under the transformation $(x_i, \sigma_{ij}) \mapsto (tx_i, t^2\sigma_{ij})$ for an arbitrary $t > 0$.

One approach to resolve the arbitrariness of the parameters introduced by such translation is to impose the constraint

$$\mathcal{C} := \frac{1}{2m} \sum_{i,j=1}^{m} s_{ij} = 1, \tag{C.7}$$

where the constant $\frac{1}{2m}$ is arbitrary, chosen for convenience as we will explain momentarily. Since $s_{ij}$ represents the variance of the difference $x_{ij} = x_i - x_j$, this constraint ensures that the total variance of all random processes $x_{ij}$ is fixed. We will further show that this constraint can be directly expressed in terms of the covariance matrix $\boldsymbol{\Sigma}$.

**Proposition C.2.** *The constraint in (C.7) is equivalent to*

$$\mathrm{trace}(\tilde{\boldsymbol{\Sigma}}) = 1, \tag{C.8}$$

*where $\tilde{\boldsymbol{\Sigma}} := \mathbf{P}\boldsymbol{\Sigma}\mathbf{P}$ is the doubly-centered covariance matrix given in (C.5), and $\mathbf{P} := \mathbf{I} - \frac{1}{m}\mathbf{1}\mathbf{1}^\intercal$ is the projection matrix defined in (B.7).*

**Proof.** Recall from (C.1) that the elements $s_{ij}$ of the matrix $\mathbf{S}$ are related to $\boldsymbol{\Sigma}$ by

$$\mathbf{S} = \mathrm{diag}(\boldsymbol{\Sigma})\mathbf{1}^\intercal + \mathbf{1}\,\mathrm{diag}(\boldsymbol{\Sigma})^\intercal - 2\boldsymbol{\Sigma}. \tag{C.9}$$

On the other hand, the constraint in (C.7) can be written equivalently as

$$\mathcal{C} = \frac{1}{2m}\mathbf{1}^\intercal \mathbf{S}\mathbf{1}. \tag{C.10}$$

Substituting (C.9) into (C.10) and noting that $\mathbf{1}^\intercal \mathrm{diag}(\boldsymbol{\Sigma}) = \mathrm{trace}(\boldsymbol{\Sigma})$, we obtain

$$\mathcal{C} = \mathrm{trace}(\boldsymbol{\Sigma}) - \frac{1}{m}\mathbf{1}^\intercal \boldsymbol{\Sigma}\mathbf{1}. \tag{C.11}$$

Next, using the cyclic property of the trace, we have $\mathrm{trace}(\tilde{\boldsymbol{\Sigma}}) = \mathrm{trace}(\mathbf{P}\boldsymbol{\Sigma}\mathbf{P}) = \mathrm{trace}(\boldsymbol{\Sigma}\mathbf{P}^2)$. Since $\mathbf{P}$ is idempotent ($\mathbf{P}^2 = \mathbf{P}$), this simplifies to

$$\mathrm{trace}(\tilde{\boldsymbol{\Sigma}}) = \mathrm{trace}(\boldsymbol{\Sigma}) - \frac{1}{m}\mathrm{trace}(\boldsymbol{\Sigma}\mathbf{1}\mathbf{1}^\intercal). \tag{C.12}$$

Moreover, by the cyclic property of the trace, we have $\mathrm{trace}(\boldsymbol{\Sigma}\mathbf{1}\mathbf{1}^\intercal) = \mathbf{1}^\intercal\boldsymbol{\Sigma}\mathbf{1}$. Substituting this into (C.12), we find that $\mathrm{trace}(\tilde{\boldsymbol{\Sigma}})$ equals $\mathcal{C}$ as expressed in (C.11). Thus, the constraint (C.7) is equivalent to $\mathrm{trace}(\tilde{\boldsymbol{\Sigma}}) = 1$, completing the proof. $\qquad\square$

By using the factor model (13), the constraint $\mathcal{C}$ can also be expressed directly in terms of the diagonal matrix $\mathbf{D}$ and the factor matrix $\boldsymbol{\Lambda}$. Additionally, for optimization purposes, we provide the derivatives of $\mathcal{C}$ with respect to these variables. The following proposition formalizes these results.

**Proposition C.3.** *If the covariance matrix $\boldsymbol{\Sigma}$ is expressed in the factorized form $\boldsymbol{\Sigma} = \mathbf{D} + \boldsymbol{\Lambda}\boldsymbol{\Lambda}^\intercal$, then the constraint $\mathcal{C}$ in (C.7) becomes*

$$\mathcal{C} = \left(1 - \frac{1}{m}\right)\mathrm{trace}(\mathbf{D}) + \|\boldsymbol{\Lambda}\|_F^2 - \frac{1}{m}\|\boldsymbol{\Lambda}^\intercal\mathbf{1}\|_2^2, \tag{C.13}$$

*where $\|\cdot\|_F$ denotes the Frobenius norm. Furthermore, the derivatives of $\mathcal{C}$ are given by*

$$\frac{\partial \mathcal{C}}{\partial \mathbf{D}} = \left(1 - \frac{1}{m}\right)\mathbf{I}, \tag{C.14a}$$

$$\frac{\partial \mathcal{C}}{\partial \boldsymbol{\Lambda}} = 2\mathbf{P}\boldsymbol{\Lambda}. \tag{C.14b}$$

**Proof.** Substituting the factorized form $\boldsymbol{\Sigma} = \mathbf{D} + \boldsymbol{\Lambda}\boldsymbol{\Lambda}^{\mathsf{T}}$ into (C.12), we have

$$\mathrm{trace}(\tilde{\boldsymbol{\Sigma}}) = \mathrm{trace}(\mathbf{D}) + \mathrm{trace}(\boldsymbol{\Lambda}\boldsymbol{\Lambda}^{\mathsf{T}}) - \frac{1}{m}\mathrm{trace}(\mathbf{D}\mathbf{1}\mathbf{1}^{\mathsf{T}}) - \frac{1}{m}\mathrm{trace}(\boldsymbol{\Lambda}\boldsymbol{\Lambda}^{\mathsf{T}}\mathbf{1}\mathbf{1}^{\mathsf{T}}). \qquad (C.15)$$

The second term simplifies as $\mathrm{trace}(\boldsymbol{\Lambda}\boldsymbol{\Lambda}^{\mathsf{T}}) = \mathrm{trace}(\boldsymbol{\Lambda}^{\mathsf{T}}\boldsymbol{\Lambda}) = \|\boldsymbol{\Lambda}\|_F^2$. For the third term, $\mathrm{trace}(\mathbf{D}\mathbf{1}\mathbf{1}^{\mathsf{T}}) = \mathbf{1}^{\mathsf{T}}\mathbf{D}\mathbf{1} = \mathrm{trace}(\mathbf{D})$ since $\mathbf{D}$ is diagonal. Also, for the fourth term, $\mathrm{trace}(\boldsymbol{\Lambda}\boldsymbol{\Lambda}^{\mathsf{T}}\mathbf{1}\mathbf{1}^{\mathsf{T}}) = \mathbf{1}^{\mathsf{T}}\boldsymbol{\Lambda}\boldsymbol{\Lambda}^{\mathsf{T}}\mathbf{1} = \|\boldsymbol{\Lambda}^{\mathsf{T}}\mathbf{1}\|_2^2$. Substituting these results into (C.15), we obtain (C.13). The derivatives with respect to $\mathbf{D}$ and $\boldsymbol{\Lambda}$ follow directly. $\qquad\square$

## APPENDIX D  EVALUATION OF STATISTICAL MODELS

In this section, we systematically evaluate the statistical models introduced in Section 2, focusing on how well they capture empirical data. We begin by detailing the experimental setup used for training and evaluating these models in Appendix D.1. This is followed by analyses of model selection, goodness of fit to observed outcomes, and generalization performance on unseen data, as detailed in Appendices D.2 to D.4, respectively. Additionally, Appendix D.5 complements the pairwise evaluations in Section 3.1 by analyzing marginal probabilities, offering an aggregated perspective on win, loss, and tie probabilities across competitors.

### D.1  DETAILS OF EXPERIMENTAL SETUP

The models used in our analysis are listed in Table D.1. Rows 1 to 6 include the Bradley-Terry (BT) model and its variants, rows 7 to 18 cover the Rao-Kupper model and its extensions, and rows 19 to 30 represent the Davidson model and its variants. For the BT model, we analyze two forms of the dataset. In rows 1 to 3, we modify the input matrices to incorporate ties by treating each tie as half a win and half a loss, i.e., $\mathbf{W} \leftarrow \mathbf{W} + \frac{1}{2}\mathbf{T}$, with row 1 following the approach by Chiang et al. (2024). In rows 4 to 6, we did not modify $\mathbf{W}$. We recall that in both cases, the BT model does not account for ties, meaning $\mathbf{T}$ is effectively ignored.

Rows 1, 4, 7, and 19, marked with ($*$) in Table D.1, represent the standard versions of the BT, Rao-Kupper, and Davidson models as found in the literature. All other rows correspond to the generalized models proposed in this work, with features detailed in the third and fourth columns of the table.

The third column represents the covariance configuration. In our analysis, we considered three cases for covariance: no covariance ($\boldsymbol{\mathsf{X}}$), $k_{\mathrm{cov}} = 0$, and 3. The symbol "$\boldsymbol{\mathsf{X}}$" indicates that the model does not include a covariance structure, excluding the Thurstonian representation. The other two cases employ the factored covariance structure $\boldsymbol{\Sigma} = \mathbf{D} + \boldsymbol{\Lambda}\boldsymbol{\Lambda}^{\mathsf{T}}$, as given in (13), where $k_{\mathrm{cov}}$ refers to the rank (i.e., the number of columns) of $\boldsymbol{\Lambda}$. The case $k_{\mathrm{cov}} = 0$ implies a diagonal covariance matrix, i.e., $\boldsymbol{\Sigma} = \mathbf{D}$, where the matrix $\boldsymbol{\Lambda}$ is absent.

The fourth column represents the tie configuration. We considered five cases: no tie ($\boldsymbol{\mathsf{X}}$), $k_{\mathrm{tie}} = 0, 1, 10$, and 20. The symbol "$\boldsymbol{\mathsf{X}}$" indicates that the model does not account for ties, while $k_{\mathrm{tie}} = 0$ corresponds to the original Rao-Kupper and Davidson models with a single tie parameter. For $k_{\mathrm{tie}} > 0$, our generalized tie model is employed, where $k_{\mathrm{tie}}$ represents the rank (i.e., the number of columns) of the matrix $\mathbf{G}$ in the factor model for ties, as defined in (6).

The fifth column represents the number of parameters in the model. All models include $m$ parameters for the score vector $\boldsymbol{x}$, where $m = 129$ corresponds to the number of competitors in the Chatbot Arena dataset. Models with a factored covariance structure include $m$ additional parameters for the diagonal matrix $\mathbf{D}$ and $m \times k_{\mathrm{cov}}$ parameters for the elements of $\boldsymbol{\Lambda}$. Similarly, models incorporating ties add one parameter $\eta$ when $k_{\mathrm{tie}} = 0$, or $m \times k_{\mathrm{tie}}$ parameters for the elements of $\mathbf{G}$ when $k_{\mathrm{tie}} > 0$.

We trained these models (except for models 1 and 4) by maximizing the likelihood function (1) using the BFGS optimization method, while satisfying the constraints in Section 2.5. This optimization method requires both the loss function $-\ell(\boldsymbol{\theta})$ and its Jacobian $-\partial\ell(\boldsymbol{\theta})/\partial\boldsymbol{\theta}$, which we analytically derived with respect to all parameters for each model and provided during training. To ensure consistency, we used a tolerance level of $\mathtt{tol} = 10^{-8}$ for convergence. Parameters were initialized as follows: scores $\boldsymbol{x}$ were initialized randomly while ensuring their sum is zero, diagonals of $\mathbf{D}$ were set to $m^{-1}$, and all other parameters were initialized to zero. Training time for each model, using an AMD EPYC 7543 processor with 32 cores, is shown in the last column of Table D.1.

| ID | Model | Model Features | | Num. Param. | $-\ell(\boldsymbol{\theta}^*)$ | Cross Entropy | | | Training Time |
|----|-------|-----|-----|-----|-----|-----|-----|-----|-----|
| | | Cov. ($k_{\text{cov}}$) | Tie ($k_{\text{tie}}$) | | | Win | Loss | Tie | |
| * 1 | Bradley-Terry | ✗ | ✗ | 129 | 0.6554 | 0.3177 | 0.3376 | — | 2.3 |
| 2 | (with tie data) | 0 | ✗ | 258 | 0.6552 | 0.3180 | 0.3371 | — | 3.8 |
| 3 | | 3 | ✗ | 645 | 0.6549 | 0.3178 | 0.3370 | — | 34.1 |
| * 4 | Bradley-Terry | ✗ | ✗ | 129 | 0.6351 | 0.3056 | 0.3295 | — | 0.0 |
| 5 | (without tie data) | 0 | ✗ | 258 | 0.6346 | 0.3059 | 0.3287 | — | 1.7 |
| 6 | | 3 | ✗ | 645 | 0.6342 | 0.3057 | 0.3285 | — | 27.5 |
| * 7 | Rao-Kupper | ✗ | 0 | 130 | 1.0095 | 0.3405 | 0.3462 | 0.3227 | 5.8 |
| 8 | | ✗ | 1 | 258 | 1.0106 | 0.3401 | 0.3459 | 0.3245 | 6.9 |
| 9 | | ✗ | 10 | 1419 | 1.0055 | 0.3404 | 0.3455 | 0.3196 | 208.1 |
| 10 | | ✗ | 20 | 2709 | 1.0050 | 0.3403 | 0.3455 | 0.3192 | 396.9 |
| 11 | | 0 | 0 | 259 | 1.0092 | 0.3408 | 0.3457 | 0.3228 | 8.4 |
| 12 | | 0 | 1 | 387 | 1.0103 | 0.3404 | 0.3454 | 0.3245 | 7.5 |
| 13 | | 0 | 10 | 1548 | 1.0052 | 0.3407 | 0.3449 | 0.3196 | 293.7 |
| 14 | | 0 | 20 | 2838 | 1.0048 | 0.3406 | 0.3449 | 0.3193 | 664.9 |
| 15 | | 3 | 0 | 646 | 1.0089 | 0.3405 | 0.3457 | 0.3227 | 36.0 |
| 16 | | 3 | 1 | 774 | 1.0100 | 0.3400 | 0.3454 | 0.3245 | 36.9 |
| 17 | | 3 | 10 | 1935 | 1.0049 | 0.3403 | 0.3449 | 0.3196 | 363.5 |
| 18 | | 3 | 20 | 3225 | 1.0044 | 0.3403 | 0.3449 | 0.3193 | 817.3 |
| *19 | Davidson | ✗ | 0 | 130 | 1.0100 | 0.3409 | 0.3461 | 0.3231 | 6.0 |
| 20 | | ✗ | 1 | 258 | 1.0077 | 0.3413 | 0.3466 | 0.3198 | 10.5 |
| 21 | | ✗ | 10 | 1419 | 1.0057 | 0.3404 | 0.3456 | 0.3197 | 253.2 |
| 22 | | ✗ | 20 | 2709 | 1.0052 | 0.3404 | 0.3455 | 0.3193 | 602.8 |
| 23 | | 0 | 0 | 259 | 1.0098 | 0.3411 | 0.3455 | 0.3231 | 8.7 |
| 24 | | 0 | 1 | 387 | 1.0074 | 0.3415 | 0.3460 | 0.3200 | 8.3 |
| 25 | | 0 | 10 | 1548 | 1.0055 | 0.3407 | 0.3451 | 0.3197 | 286.9 |
| 26 | | 0 | 20 | 2838 | 1.0050 | 0.3407 | 0.3450 | 0.3194 | 665.1 |
| 27 | | 3 | 0 | 646 | 1.0094 | 0.3410 | 0.3453 | 0.3231 | 34.6 |
| 28 | | 3 | 1 | 774 | 1.0070 | 0.3412 | 0.3460 | 0.3199 | 35.8 |
| 29 | | 3 | 10 | 1935 | 1.0051 | 0.3407 | 0.3448 | 0.3197 | 366.4 |
| 30 | | 3 | 20 | 3225 | 1.0047 | 0.3405 | 0.3448 | 0.3194 | 804.9 |

Table D.1: Configurations and training details of the 30 statistical models used throughout the analysis. These models are referenced by their ID in various sections of the paper. Rows marked with (∗) represent the prior models, while unmarked rows correspond to the generalized models proposed in this work.

Models in rows 1 and 4 were trained using the iterative minorization–maximization (MM) algorithm of Newman (2023), which offers notable speed advantages over conventional maximum likelihood estimation. MM algorithms have also been extended to certain generalizations of the Bradley-Terry model, as shown by Hunter (2004). Whether MM methods are directly applicable to the more complex generalized models proposed in this work remains an open question and warrants further investigation.

To verify the robustness of our approach, we tested global minimization methods, including basin-hopping (Wales & Doye, 1997) and simplicial homology global optimization (SHGO) (Endres et al., 2018). While these global methods required significantly longer computation times, they produced solutions consistent with those found by BFGS, confirming the reliability of local optimization for this problem.

## D.2 MODEL SELECTION

To evaluate model fit, we computed the negative log-likelihood (NLL) and the cross-entropy losses for win, loss, and tie outcomes, as shown in the sixth to ninth columns of Table D.1.

The cross-entropy loss quantifies how well the predicted probabilities align with the empirical probabilities derived from observed data. For a given pair $\{i, j\}$, the empirical probabilities are computed

as

$$P^e(i \succ j \mid \{i,j\}) = \frac{w_{ij}}{n_{ij}},$$

$$P^e(i \prec j \mid \{i,j\}) = \frac{w_{ji}}{n_{ij}},$$

$$P^e(i \sim j \mid \{i,j\}) = \frac{t_{ij}}{n_{ij}},$$

where $n_{ij} := w_{ij} + w_{ji} + t_{ij}$ is the total number of matches between competitors $i$ and $j$. The overall cross-entropy losses, $H$, for win, loss, and tie outcomes are calculated as

$$H_{\text{win}}(\boldsymbol{\theta}) = - \sum_{\{i,j\} \in E} P^e(i \succ j \mid \{i,j\}) \log \left( P(i \succ j \mid \{i,j\})(\boldsymbol{\theta}) \right) P(\{i,j\} \mid E), \quad \text{(D.2a)}$$

$$H_{\text{loss}}(\boldsymbol{\theta}) = - \sum_{\{i,j\} \in E} P^e(i \prec j \mid \{i,j\}) \log \left( P(i \prec j \mid \{i,j\})(\boldsymbol{\theta}) \right) P(\{i,j\} \mid E), \quad \text{(D.2b)}$$

$$H_{\text{tie}}(\boldsymbol{\theta}) = - \sum_{\{i,j\} \in E} P^e(i \sim j \mid \{i,j\}) \log \left( P(i \sim j \mid \{i,j\})(\boldsymbol{\theta}) \right) P(\{i,j\} \mid E), \quad \text{(D.2c)}$$

where $P(\{i,j\} \mid E)$ is the probability of observing a match for the pair $\{i,j\}$, given by

$$P(\{i,j\} \mid E) = \frac{n_{ij}}{n}, \quad \text{where} \quad n := \sum_{\{k,l\} \in E} n_{kl}. \quad \text{(D.3)}$$

The cross-entropy losses for win, loss, and tie outcomes at $\boldsymbol{\theta}^*$, the optimal parameter values, are reported in the seventh, eighth, and ninth columns of Table D.1, respectively.

The sixth column reports the negative log-likelihood (NLL) of the models at $\boldsymbol{\theta}^*$, where

$$\ell(\boldsymbol{\theta}) = \frac{1}{n} \sum_{\{i,j\} \in E} w_{ij} \log \left( P_{i \succ j}(\boldsymbol{\theta}) \right) + w_{ji} \log \left( P_{i \prec j}(\boldsymbol{\theta}) \right) + t_{ij} \log \left( P_{i \sim j}(\boldsymbol{\theta}) \right). \quad \text{(D.4)}$$

This log-likelihood slightly differs from the expression in (1), as it excludes a constant term and is scaled by $\frac{1}{n}$, where $n$ is defined in (D.3). Notably, the NLL here is equivalent to the sum of the cross-entropy losses for win, loss, and tie outcomes:

$$-\ell(\boldsymbol{\theta}) = H_{\text{win}}(\boldsymbol{\theta}) + H_{\text{loss}}(\boldsymbol{\theta}) + H_{\text{tie}}(\boldsymbol{\theta}).$$

This relationship can be verified by summing the values in the seventh, eighth, and ninth columns to obtain the value in the sixth column.

Since the BT models do not account for ties, their NLL values are lower than those of tie-inclusive models. This is because the NLL of BT models includes only two terms, $H_{\text{win}}$ and $H_{\text{loss}}$, while tie-inclusive models also include $H_{\text{tie}}$. Consequently, direct comparison of NLLs between these model types is not meaningful.

On the other hand, when comparing individual cross-entropy losses, $H_{\text{win}}$ and $H_{\text{loss}}$, we observe that the BT models achieve lower values compared to non-BT models. This discrepancy arises because the BT models predict two outcomes (win and loss), yielding only one independent output, as the probability of loss complements the probability of a win. In contrast, the Rao-Kupper and Davidson models predict three outcomes (win, loss, tie), resulting in two independent output variables. Thus, the BT models fit a *one-dimensional* output space, while the other models fit a *two-dimensional* space. Although the BT models achieve lower cross-entropy losses, the complexity of the Rao-Kupper and Davidson models offers richer predictions.

Within each model category, we observe that increasing the rank $k_{\text{cov}}$ for covariance or $k_{\text{tie}}$ for tie models consistently improves the NLL and cross-entropies, indicating better fit. Further evaluation metrics, including goodness-of-fit and generalization performance, are provided in Appendix D.3 and Appendix D.4, respectively.

## D.3  MODEL FIT

To assess model fit, we computed multiple metrics, including root-mean-square error (RMSE), Kullback-Leibler (KL) divergence, and Jensen-Shannon (JS) divergence, as presented in Table D.2.

| ID | Model | Model Features | | RMSE | | | | Divergence ($\times 10^2$) | |
| --- | --- | --- | --- | --- | --- | --- | --- | --- | --- |
| | | Cov. ($k_{\mathrm{cov}}$) | Tie ($k_{\mathrm{tie}}$) | Win | Loss | Tie | All | KLD | JSD |
| * 1 | Bradley-Terry | ✗ | ✗ | 29.7 | 29.7 | — | 29.7 | 1.49 | 0.44 |
| 2 | (*with tie data*) | 0 | ✗ | 26.2 | 26.2 | — | 26.2 | 1.42 | 0.42 |
| 3 | | 3 | ✗ | 17.4 | 17.4 | — | 17.4 | 1.30 | 0.39 |
| * 4 | Bradley-Terry | ✗ | ✗ | 35.1 | 35.1 | — | 35.1 | 1.82 | 0.52 |
| 5 | (*without tie data*) | 0 | ✗ | 31.5 | 31.5 | — | 31.5 | 1.71 | 0.49 |
| 6 | | 3 | ✗ | 17.3 | 17.3 | — | 17.3 | 1.58 | 0.46 |
| * 7 | Rao-Kupper | ✗ | 0 | 48.2 | 69.9 | 103.5 | 77.3 | 3.32 | 0.92 |
| 8 | | ✗ | 1 | 46.4 | 67.8 | 99.2 | 74.3 | 3.45 | 0.91 |
| 9 | | ✗ | 10 | 34.1 | 34.2 | 23.1 | 30.9 | 2.63 | 0.73 |
| 10 | | ✗ | 20 | 34.3 | 32.2 | 16.8 | 28.8 | 2.35 | 0.65 |
| 11 | | 0 | 0 | 46.5 | 67.9 | 103.6 | 76.4 | 3.23 | 0.90 |
| 12 | | 0 | 1 | 43.5 | 66.8 | 99.4 | 73.5 | 3.36 | 0.89 |
| 13 | | 0 | 10 | 29.8 | 31.6 | 22.7 | 28.3 | 2.55 | 0.70 |
| 14 | | 0 | 20 | 30.4 | 29.1 | 16.7 | 26.1 | 2.26 | 0.63 |
| 15 | | 3 | 0 | 49.0 | 61.7 | 104.7 | 75.6 | 3.09 | 0.86 |
| 16 | | 3 | 1 | 48.6 | 58.7 | 100.9 | 73.0 | 3.18 | 0.84 |
| 17 | | 3 | 10 | 20.0 | 21.2 | 22.1 | 21.1 | 2.42 | 0.67 |
| 18 | | 3 | 20 | 18.7 | 18.9 | 15.8 | 17.9 | 2.12 | 0.59 |
| *19 | Davidson | ✗ | 0 | 51.0 | 71.8 | 109.8 | 81.3 | 3.41 | 0.94 |
| 20 | | ✗ | 1 | 44.4 | 63.3 | 90.1 | 68.6 | 2.99 | 0.82 |
| 21 | | ✗ | 10 | 37.1 | 39.6 | 25.7 | 34.7 | 2.69 | 0.75 |
| 22 | | ✗ | 20 | 37.7 | 37.1 | 17.2 | 32.1 | 2.50 | 0.70 |
| 23 | | 0 | 0 | 49.4 | 70.5 | 109.9 | 80.6 | 3.32 | 0.92 |
| 24 | | 0 | 1 | 41.1 | 62.4 | 91.4 | 68.1 | 2.94 | 0.81 |
| 25 | | 0 | 10 | 32.8 | 37.7 | 27.0 | 32.8 | 2.73 | 0.76 |
| 26 | | 0 | 20 | 35.7 | 32.6 | 18.8 | 30.0 | 2.56 | 0.72 |
| 27 | | 3 | 0 | 55.1 | 61.1 | 111.0 | 79.8 | 3.18 | 0.89 |
| 28 | | 3 | 1 | 46.5 | 50.0 | 90.6 | 65.5 | 2.80 | 0.78 |
| 29 | | 3 | 10 | 20.8 | 22.0 | 25.0 | 22.7 | 2.57 | 0.72 |
| 30 | | 3 | 20 | 19.1 | 19.0 | 17.1 | 18.4 | 2.43 | 0.68 |

Table D.2: Goodness-of-fit metrics, including root-mean-square error (RMSE), Kullback-Leibler divergence (KLD), and Jensen-Shannon divergence (JSD), for the 30 statistical models introduced in Table D.1. Rows marked with (∗) represent the prior models, while unmarked rows correspond to the generalized models proposed in this work.

**Root-Mean-Square Error (RMSE).** The RMSE quantifies the deviation between the predicted and observed match frequencies. For each pair $\{i, j\} \in E$, the predicted frequencies for win, loss, and tie outcomes are computed as:

$$\hat{w}_{ij} := n_{ij} P_{i \succ j}, \quad \hat{w}_{ji} := n_{ij} P_{i \prec j}, \quad \hat{t}_{ij} := n_{ij} P_{i \sim j}.$$

The weighted squared errors are given by

$$e_{\mathrm{win}}^2 = \sum_{\{i,j\} \in E} \frac{n_{ij}}{n} (w_{ij} - \hat{w}_{ij})^2,$$

$$e_{\mathrm{loss}}^2 = \sum_{\{i,j\} \in E} \frac{n_{ij}}{n} (w_{ji} - \hat{w}_{ji})^2,$$

$$e_{\mathrm{tie}}^2 = \sum_{\{i,j\} \in E} \frac{n_{ij}}{n} (t_{ij} - \hat{t}_{ij})^2.$$

The above errors are weighted by $n_{ij}/n$, the probability of observing a match for the pair $\{i, j\}$, as given in (D.3), to account for imbalances in the number of matches per pair. The square roots of these quantities ($e_{\mathrm{win}}, e_{\mathrm{loss}}, e_{\mathrm{tie}}$) are reported in the fifth to seventh columns of Table D.2. The overall

RMSE, $e_{\text{all}}$, is shown in the eighth column, where $e_{\text{all}}^2$ is defined as the average of $e_{\text{win}}^2$, $e_{\text{loss}}^2$, and $e_{\text{tie}}^2$. Similar to earlier trends, BT models show lower errors due to their simpler output space. However, in each category, models with higher $k_{\text{cov}}$ values for covariance and $k_{\text{tie}}$ for ties demonstrate notable improvements in model accuracy.

**Divergence Metrics.** To further evaluate model fit, we computed the KL and JS divergences between the predicted and empirical probability distributions. To define divergences, we treated the three outcomes of win, loss, and tie as classes of discrete probability mass functions with $P_{ij}^e := (P_{i \succ j}^e, P_{i \prec j}^e, P_{i \sim j}^e)$ and $P_{ij} := (P_{i \succ j}, P_{i \prec j}, P_{i \sim j})$. The KL divergence is defined as

$$
D_{\text{KL}}(P^e \| P) = \frac{1}{|E|} \sum_{\{i,j\} \in E} P_{i \succ j}^e \log\left(\frac{P_{i \succ j}^e}{P_{i \succ j}}\right) + P_{i \prec j}^e \log\left(\frac{P_{i \prec j}^e}{P_{i \prec j}}\right) + P_{i \sim j}^e \log\left(\frac{P_{i \sim j}^e}{P_{i \sim j}}\right).
$$

Accordingly, the JS divergence is defined by

$$
D_{\text{JS}}(P^e \| P) := \frac{1}{2} \left( D_{\text{KL}}(P^e \| M) + D_{\text{KL}}(P \| M) \right),
$$

where for each pair $\{i, j\}$, we defined the mixture distribution $M_{ij} := \frac{1}{2}(P_{ij}^e + P_{ij})$. The JS divergence is symmetric and bounded between 0 and 1, making it intuitive for interpretation.

The KL and JS divergences are shown in the ninth and tenth columns of Table D.2. Lower KL and JS values indicate better model fit. Notably, models incorporating covariance and tie factor models yield better results in terms of divergence, reaffirming the effectiveness of these extensions.

## D.4 GENERALIZATION PERFORMANCE

To evaluate the models' generalization performance, we trained each model on $90\%$ of the data and tested predictions on the remaining $10\%$, with the data randomly split into training and test sets. Results for the weighted RMSE are presented in the fifth to eighth columns of Table D.3, while the KL and JS divergences are shown in the ninth and tenth columns, respectively. The definitions of RMSE, KL divergence, and JS divergence remain the same as those in Appendix D.3, but here, these metrics are computed on the test data rather than the training data. These generalization results complement the training results presented in Table D.2, highlighting how model performance extends to unseen data.

These metrics reveal that increasing the number of parameters, such as $k_{\text{cov}}$ for covariance or $k_{\text{tie}}$ for tie factor models, initially enhances generalization performance. However, beyond a certain threshold, additional parameters lead to diminishing returns in generalization, even as training fit improves (Table D.2). This phenomenon is attributed to overparameterization, where the model becomes too complex relative to the amount of data available.

The ratio of model parameters (fifth column of Table D.1) to the number of data points ($|E| = 3455$, the number of pairs) provides insight into this overparameterization. Models with $k_{\text{tie}} = 1$ to $k_{\text{tie}} = 10$ maintain a balanced parameter-to-data ratio, achieving strong generalization and fit, as reflected in the RMSE values in Table D.3. In contrast, models with $k_{\text{tie}} = 20$ approach the upper limit of this ratio, resulting in improved training fit but reduced generalization, signaling the onset of overfitting.

These results highlight the importance of balancing model complexity with data availability to achieve optimal generalization performance.

## D.5 EVALUATING MARGINAL PROBABILITIES OF WIN, LOSS, AND TIE

The errors in previous sections were calculated based on pairwise probabilities, such as $P_{i \succ j}$, with errors averaged over all pairwise comparisons in $E$. Here, we assess the *marginal* probabilities for each competitor, which represent the overall likelihood of winning, losing, or tying against all other competitors. Specifically, we denote these probabilities respectively by $P(i \succ V \setminus \{i\} \,|\, E)$,

| ID | Model | Model Features | | RMSE | | | | Divergence ($\times 10^2$) | |
|---|---|---|---|---|---|---|---|---|---|
| | | Cov. ($k_{\text{cov}}$) | Tie ($k_{\text{tie}}$) | Win | Loss | Tie | All | KLD | JSD |
| ∗ 1 | Bradley-Terry | ✗ | ✗ | 27.4 | 27.4 | — | 27.4 | 1.46 | 0.41 |
| 2 | (*with tie data*) | 0 | ✗ | 27.6 | 27.6 | — | 27.6 | 1.48 | 0.41 |
| 3 | | 3 | ✗ | 27.1 | 27.1 | — | 27.1 | 1.55 | 0.43 |
| ∗ 4 | Bradley-Terry | ✗ | ✗ | 30.0 | 30.0 | — | 30.0 | 1.74 | 0.48 |
| 5 | (*without tie data*) | 0 | ✗ | 30.3 | 30.3 | — | 30.3 | 1.77 | 0.48 |
| 6 | | 3 | ✗ | 30.4 | 30.4 | — | 30.4 | 2.06 | 0.54 |
| ∗ 7 | Rao-Kupper | ✗ | 0 | 54.5 | 29.1 | 67.4 | 52.8 | 3.16 | 0.88 |
| 8 | | ✗ | 1 | 49.9 | 41.9 | 74.3 | 57.0 | 3.31 | 0.87 |
| 9 | | ✗ | 10 | 26.3 | 38.6 | 32.1 | 32.7 | 3.17 | 0.85 |
| 10 | | ✗ | 20 | 29.0 | 56.3 | 66.4 | 53.0 | 4.14 | 1.00 |
| 11 | | 0 | 0 | 52.9 | 31.5 | 67.8 | 52.9 | 3.19 | 0.88 |
| 12 | | 0 | 1 | 46.8 | 45.5 | 75.0 | 57.4 | 3.31 | 0.87 |
| 13 | | 0 | 10 | 25.9 | 38.5 | 31.6 | 32.4 | 3.10 | 0.84 |
| 14 | | 0 | 20 | 30.2 | 54.3 | 64.6 | 51.7 | 4.66 | 1.06 |
| 15 | | 3 | 0 | 50.3 | 33.8 | 68.1 | 52.6 | 3.31 | 0.90 |
| 16 | | 3 | 1 | 51.6 | 42.1 | 75.0 | 57.9 | 3.59 | 0.93 |
| 17 | | 3 | 10 | 28.5 | 34.9 | 32.3 | 32.0 | 3.25 | 0.87 |
| 18 | | 3 | 20 | 35.9 | 59.6 | 67.1 | 55.8 | 4.71 | 1.07 |
| ∗19 | Davidson | ✗ | 0 | 54.7 | 30.8 | 70.0 | 54.3 | 3.28 | 0.91 |
| 20 | | ✗ | 1 | 43.1 | 24.6 | 42.7 | 37.8 | 2.76 | 0.77 |
| 21 | | ✗ | 10 | 27.6 | 41.8 | 31.4 | 34.2 | 2.95 | 0.81 |
| 22 | | ✗ | 20 | 28.6 | 65.5 | 73.4 | 59.1 | 3.42 | 0.93 |
| 23 | | 0 | 0 | 54.0 | 32.8 | 70.2 | 54.5 | 3.31 | 0.92 |
| 24 | | 0 | 1 | 44.2 | 26.2 | 45.0 | 39.4 | 2.91 | 0.81 |
| 25 | | 0 | 10 | 26.9 | 40.0 | 28.6 | 32.3 | 3.06 | 0.84 |
| 26 | | 0 | 20 | 31.0 | 68.3 | 80.4 | 63.5 | 3.51 | 0.96 |
| 27 | | 3 | 0 | 52.5 | 34.2 | 70.2 | 54.3 | 3.40 | 0.93 |
| 28 | | 3 | 1 | 40.7 | 28.2 | 44.0 | 38.2 | 2.87 | 0.79 |
| 29 | | 3 | 10 | 33.6 | 32.7 | 28.5 | 31.6 | 3.31 | 0.89 |
| 30 | | 3 | 20 | 32.8 | 71.6 | 83.3 | 66.2 | 3.70 | 1.00 |

Table D.3: Generalization performance of the 30 statistical models introduced in Table D.1, evaluated on test data using a 90/10 train-test split, including root-mean-square error (RMSE), Kullback-Leibler divergence (KLD), and Jensen-Shannon divergence (JSD). Rows marked with (∗) represent the prior models, while unmarked rows correspond to the generalized models proposed in this work.

$P(i \prec V \setminus \{i\} \,|\, E)$, and $P(i \sim V \setminus \{i\} \,|\, E)$, which are defined by

$$P(i \succ V \setminus \{i\} \,|\, E) = \sum_{\{i,j\} \in E(i)} P(i \succ j \,|\, \{i,j\}) \, P(\{i,j\} \,|\, E), \tag{D.5a}$$

$$P(i \prec V \setminus \{i\} \,|\, E) = \sum_{\{i,j\} \in E(i)} P(i \prec j \,|\, \{i,j\}) \, P(\{i,j\} \,|\, E), \tag{D.5b}$$

$$P(i \sim V \setminus \{i\} \,|\, E) = \sum_{\{i,j\} \in E(i)} P(i \sim j \,|\, \{i,j\}) \, P(\{i,j\} \,|\, E), \tag{D.5c}$$

where $E(i) := \{e \in E \,|\, i \in e\}$ represents the set of edges incident to the vertex $i \in V$, and $P(\{i,j\} \,|\, E)$ is the probability of observing a match for the pair $\{i,j\}$, as given by (D.3). For brevity, we denote the marginal probabilities in (D.5a) to (D.5c) by $P_{i \succ V_{-i}}$, $P_{i \prec V_{-i}}$, and $P_{i \sim V_{-i}}$, respectively, where $V_{-i} := V \setminus \{i\}$.

Here, we evaluate the goodness of fit of the models by comparing the predicted marginal probabilities of winning, losing, and tying for each competitor against their corresponding empirical marginal probabilities. Figure D.1 illustrates the marginal probabilities for a selected set of models. Specifically, the first two rows of the figure show results from the BT models (Models 1 and 4 of Table D.1,

with and without modified input data), while rows 3 to 5 correspond to the Rao-Kupper models (Model 7 as the standard model with one tie parameter corresponding to $k_{\text{tie}} = 0$, Model 8 with factored tie model and $k_{\text{tie}} = 1$, and Model 10 with factored tie model and $k_{\text{tie}} = 20$). Results for the Davidson models are omitted as they closely resemble those of the Rao-Kupper models under similar conditions.

The left, middle, and right columns in the figure show the marginal probabilities of win ($P_{i \succ V_{-i}}$), loss ($P_{i \prec V_{-i}}$), and tie ($P_{i \sim V_{-i}}$), respectively. Each row shares the same legend, shown only in the rightmost column. The abscissa represent competitors ordered by their rank in Model 18. The colored curves represent predicted marginal probabilities for each model, with red-themed curves for BT models and green-themed curves for Rao-Kupper models. The black curve represents the empirical marginal probabilities from the observed data, though in some cases, it may overlap with the colored curves. The relative error between model predictions and empirical probabilities is presented in the sixth row, using the same color scheme for consistency. Key observations are as follows:

First, in the top two rows of the figure, the BT model predictions (red curves) noticeably deviate from the empirical probabilities (black curve). This is because the BT models do not account for ties, resulting in a different distribution of win and loss probabilities. To provide a fair comparison, we compare BT model predictions with adjusted empirical probabilities, represented by the dotted black curve, which excludes ties. Accordingly, the relative error for BT models in the sixth row is computed against these adjusted probabilities. By contrast, Rao-Kupper models are compared directly with empirical probabilities from the input data, which include ties.

Second, in the last row of the figure, we observe that the errors for BT models are generally lower than those of the Rao-Kupper models. This is due to the smaller output dimension of the BT models, which fit only win and loss outcomes. If BT models were compared to the actual empirical probabilities (solid black curves), their error would be higher than that of the Rao-Kupper models. However, such a comparison would be unfair, as BT models are trained on modified data that does not account for ties.

Finally, the original Rao-Kupper model with a single tie parameter (Model 7, shown in the third row) exhibits errors on the scale of 1 to 10, making it impractical for real-world applications, particularly in predicting ties. However, our generalized Rao-Kupper models, which incorporate factored tie models (Models 8 and 10, shown in the fourth and fifth rows), demonstrate a substantial improvement in accuracy. This enhancement not only elevates the prediction of ties but also improves the prediction of win and loss outcomes by one to two orders of magnitude. This result is significant, as it brings the Rao-Kupper and Davidson models back into practical relevance, offering richer predictions for win, loss, and tie outcomes—unlike the BT models—without compromising on accuracy.

## APPENDIX E    COMPARATIVE ANALYSIS OF RANKING VARIABILITY

This section expands on the ranking comparisons presented in Section 4, offering a detailed analysis of how rankings vary across statistical models with different parameter configurations. Visualizations include a representative model's score distribution (Appendix E.1), a comparison of rankings across models (Appendix E.2), and Kendall's $\tau$ correlation matrix (Appendix E.3) to assess ranking similarities. Hierarchical clustering (Appendix E.4) further uncovers structural patterns in model-based rankings, highlighting the distinct effects of covariance and tie parameters on ranking alignment.

### E.1    REPRESENTATIVE MODEL RANKINGS AND SCORE DISTRIBUTION

The score plot in Figure E.1 illustrates the rankings and scores of chatbots, using Model 18 from Table D.1 as an example. This generalized Rao-Kupper (RK) model incorporates covariance ($k_{\text{cov}} = 3$) and tie factor parameters ($k_{\text{tie}} = 20$), achieving strong goodness-of-fit metrics. Each bar represents a chatbot, ordered from the highest to the lowest score, showcasing their relative performance.

Other models with similar configurations, such as the generalized Davidson model (e.g., Model 30), produce comparable score distributions and rankings. Appendix E.2 examines ranking similarity and variability across a selection of models with diverse parameterizations.

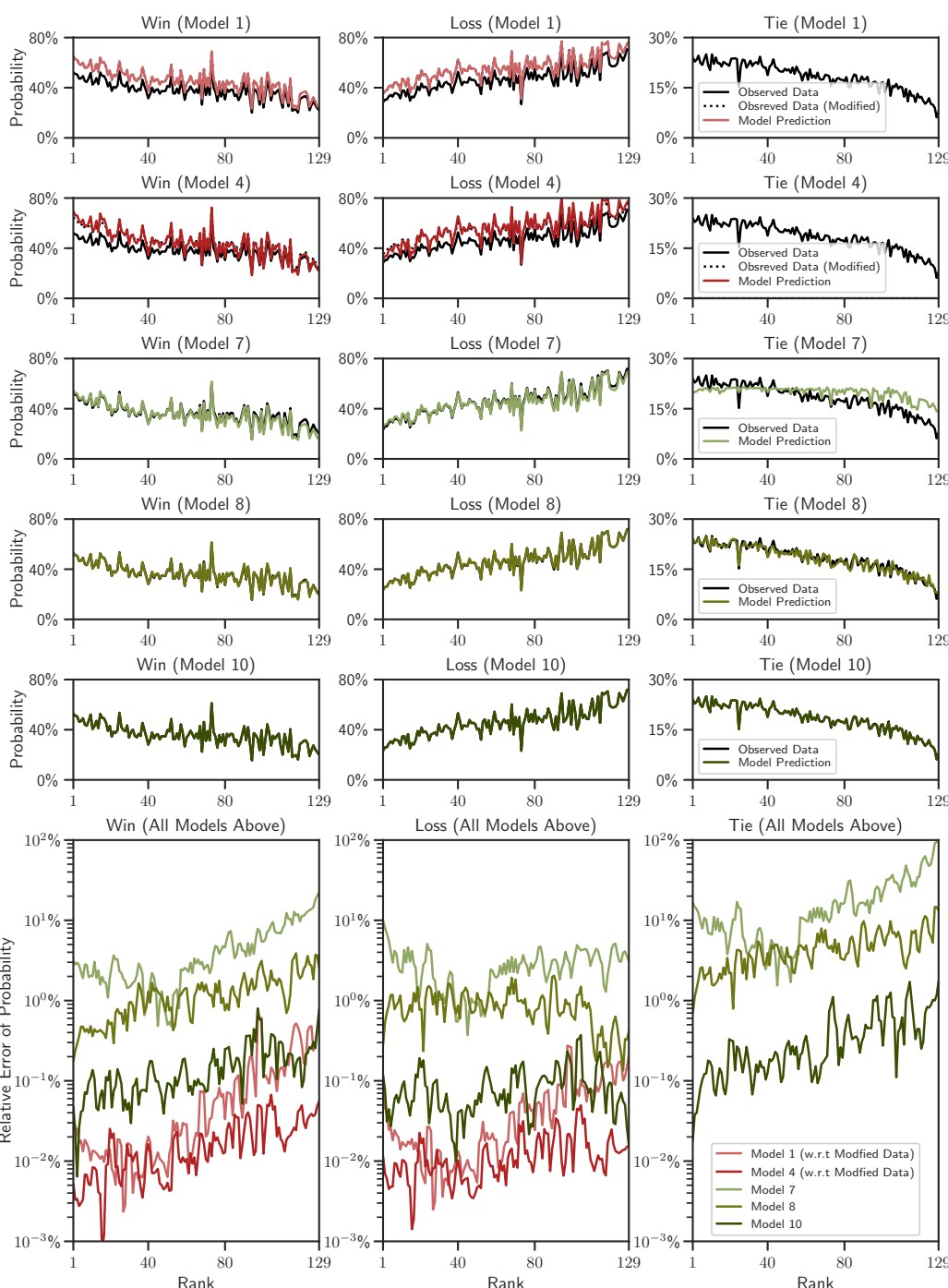

Figure D.1: Comparison of predicted (colored curves) and empirical (black curves) marginal probabilities of win (left), loss (middle), and tie (right) for selected models. First and second rows: BT models, third row: original Rao-Kupper with tie factor $k_{\text{tie}} = 0$, fourth and fifth rows: generalized Rao-Kupper with tie factors $k_{\text{tie}} = 1$ and $k_{\text{tie}} = 20$. Sixth row: relative errors for rows one to five, calculated between predicted and empirical probabilities.

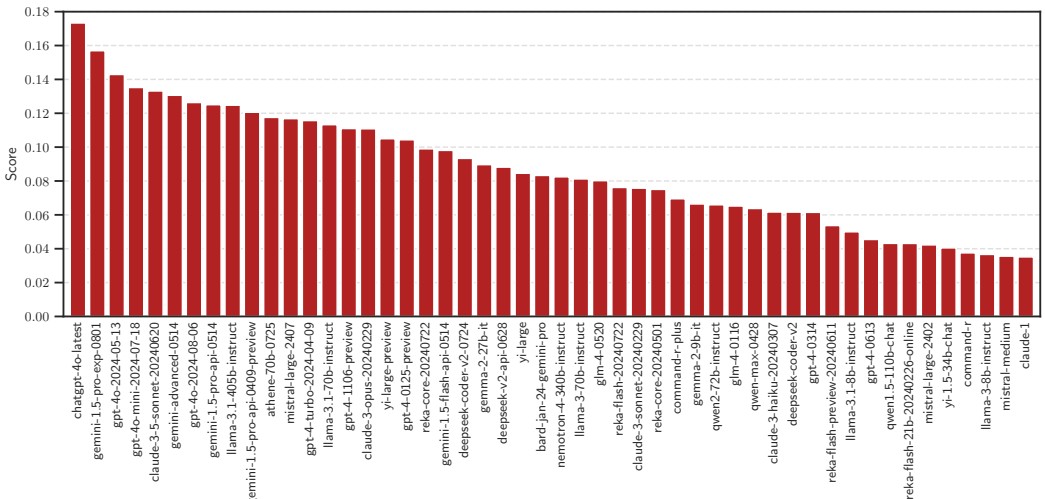

Figure E.1: Competitors ranked by their scores according to Model 18 from Table D.1.

## E.2 EXPLORING RANKING SIMILARITY ACROSS MODELS

The bump chart in Figure E.2 visualizes how rankings vary across 12 selected models from Table D.1. Each column represents a model, arranged from simpler models on the right to more complex ones on the left, while each row tracks a chatbot's rank across models.

To comprehensively explore ranking similarity, we selected 12 models with diverse configurations. These include baseline models without covariance or tie factor generalizations (Models 1, 4, 7, and 19), models incorporating covariance ($k_{\text{cov}} = 0$, Models 3 and 6), models introducing tie factor parameters ($k_{\text{tie}} = 20$, Models 10 and 22), and the most complex models combining both covariance ($k_{\text{cov}} = 3$) and tie factors ($k_{\text{tie}} = 20$, Models 18 and 30).

The rankings of top competitors remain consistent across all models, reflecting stable performance at the top. In contrast, discrepancies grow more pronounced at lower ranks, where models with additional parameters for ties and covariances produce differing results. Notably, similar models—especially those with comparable configurations—tend to yield similar rankings, with occasional exceptions. A detailed analysis of ranking similarities is provided in Appendix E.3.

## E.3 QUANTIFYING RANKING SIMILARITY ACROSS MODELS

While the bump chart in Appendix E.2 provides a visual overview of ranking shifts across models, quantifying the degree of similarity between these rankings requires statistical correlation measures. A variety of methods are available, including Pearson's correlation, Spearman's $\rho$, and Kendall's $\tau$ (Kendall, 1938). Pearson's correlation is most suitable for assessing linear relationships between continuous variables, while Spearman's rank correlation generalizes it for monotonic relationships in ordinal data. However, neither offers as direct an interpretation for pairwise ranking comparisons as Kendall's $\tau$, which evaluates the ordering of pairs directly, making it particularly well-suited for ordinal data.

Kendall's ranking correlation quantifies the agreement between two ranking orders by comparing the relative ordering of pairs of objects. Given two score vectors, $\boldsymbol{x}^p = (x_1^p, \ldots, x_m^p)$ and $\boldsymbol{x}^q = (x_1^q, \ldots, x_m^q)$, from the $p$-th and $q$-th models, respectively, $\tau_{pq}$ reflects the extent to which the pairwise orderings are concordant. Two pairs, $(x_i^p, x_j^p)$ and $(x_i^q, x_j^q)$, are *concordant* if they maintain the same relative ordering—i.e., either $x_i^p < x_j^p$ and $x_i^q < x_j^q$, or $x_i^p > x_j^p$ and $x_i^q > x_j^q$. Conversely, they are *discordant* if the orderings are reversed. This concordance criterion can be expressed as $\mathsf{sgn}(x_i^p - x_j^p)\,\mathsf{sgn}(x_i^q - x_j^q) = 1$ for concordant pairs, and $-1$ for discordant pairs.

The Kendall correlation $\tau_{pq}$ is defined as the difference between the probabilities of concordant and discordant pairs and can be empirically obtained by computing the difference of counts for all

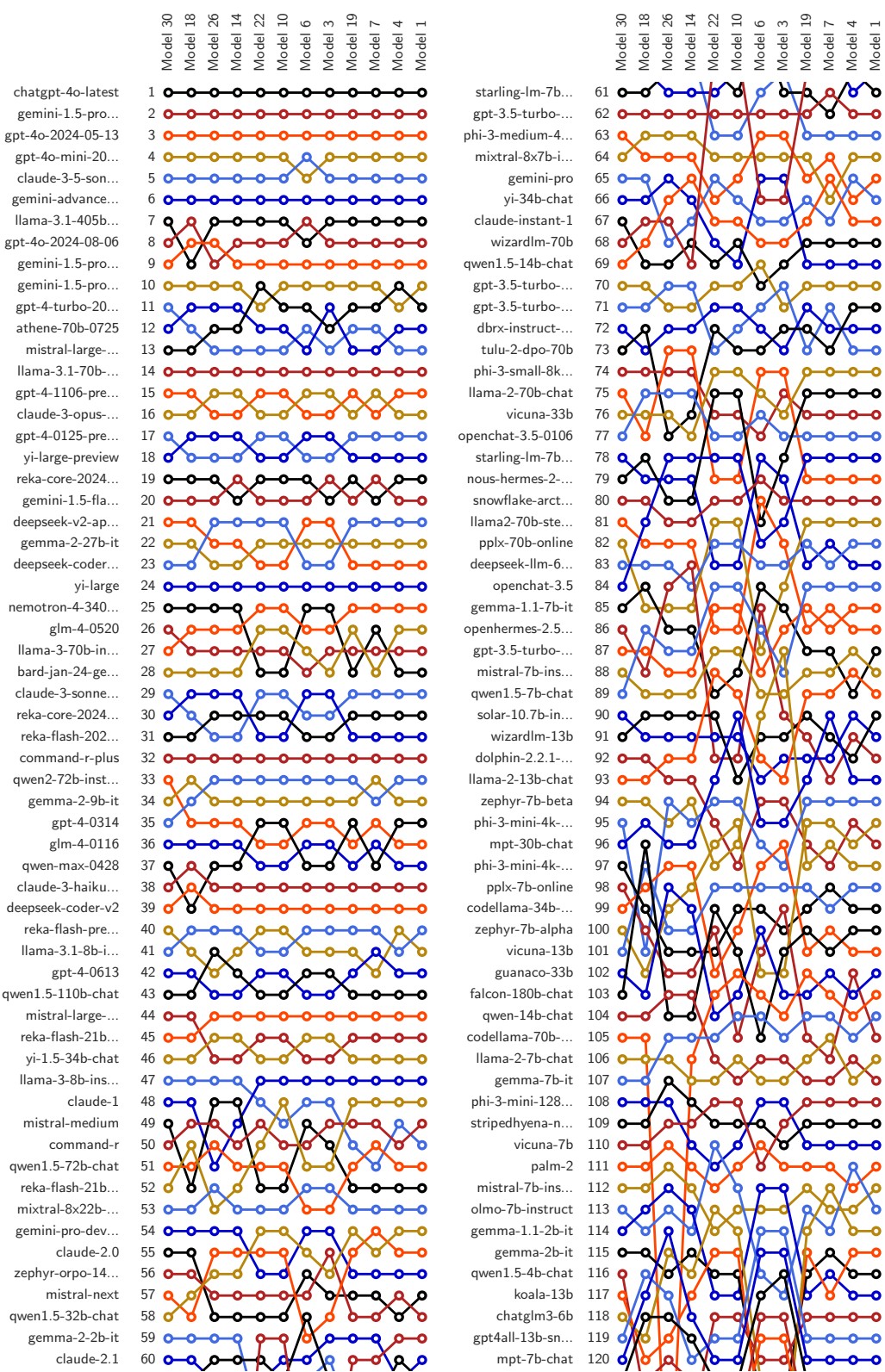

Figure E.2: Bump chart comparing chatbot rankings across 12 statistical models, with Model 1 representing the Elo-based ranking method used in Chiang et al. (2024). Models are arranged with increasing complexity from right to left. Lines track changes in ranking for each chatbot.

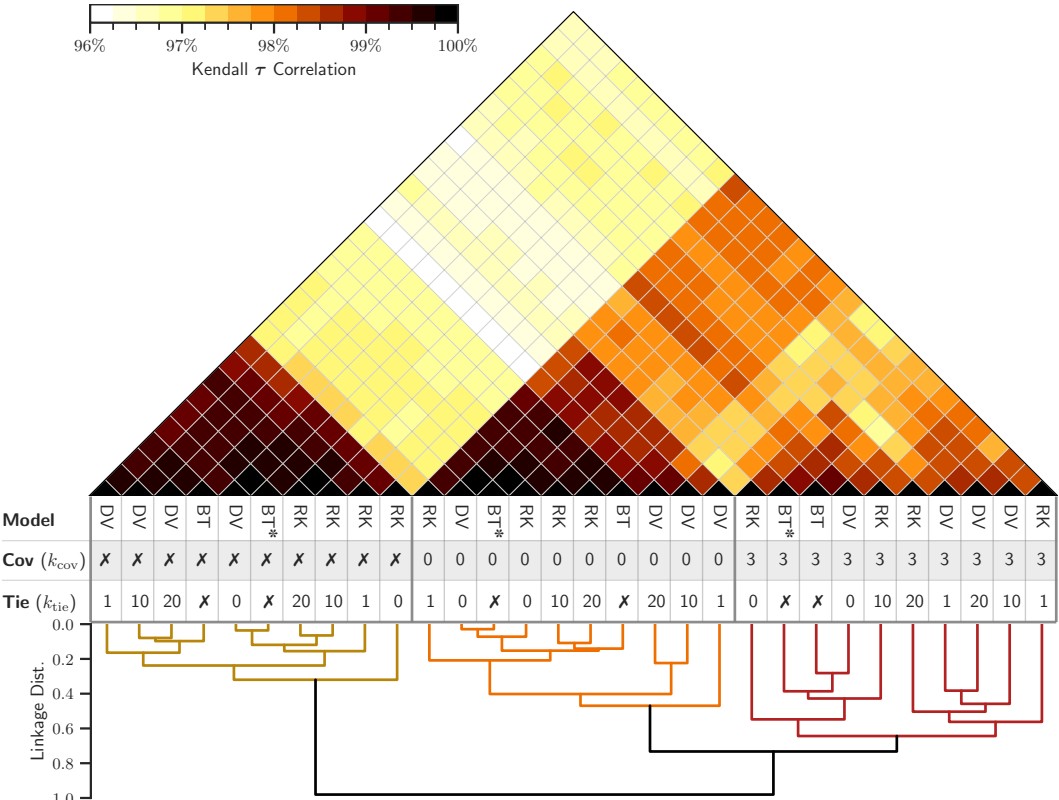

Figure E.3: Kendall $\tau$ ranking correlation matrix for models in Table D.1. The table below categorizes models by configuration: model type (first row), covariance factor $k_{\text{cov}}$ (second row), and tie factor $k_{\text{tie}}$ (third row). In the first row, code names BT, RK, and DV denote Bradley-Terry, Rao-Kupper, and Davidson models, respectively, with BT$^*$ indicating the Bradley-Terry model treating ties as half wins and half losses. The model order is determined by hierarchical clustering on Kendall's correlation values, highlighting two main clusters based on the presence of covariance, with a further division within the covariance group based on factor $k_{\text{cov}}$. A dendrogram below the table illustrates this clustering.

concordant and discordant pairs, normalized by the total number of pairs, $\binom{m}{2}$, as

$$\tau_{pq} = \frac{1}{\binom{m}{2}} \sum_{1 \le i < j \le m} \text{sgn}(x_i^p - x_j^p) \, \text{sgn}(x_i^q - x_j^q). \tag{E.1}$$

This correlation ranges from $-1$ to $1$, where $\tau_{pq} = 1$ indicates identical rankings, and $\tau_{pq} = -1$ implies a complete reversal in ranking order (i.e., $x_i^p < x_j^p$ implies $x_i^q > x_j^q$ and vice versa). The probability that a pairwise order $x_i^p < x_j^p$ in one ranking aligns with $x_i^q < x_j^q$ in another is $\frac{1}{2}(\tau_{pq} + 1)$ (Gibbons & Chakraborti, 2003, p. 410).

In this analysis, we compute the Kendall correlation matrix $\boldsymbol{\tau} = [\tau_{pq}]$, $p, q = 1, \ldots, 30$, between each pair of models in Table D.1 using Kendall's $\tau$-b method, which also accounts for ties in the scores (Kendall, 1945).

Figure E.3 shows the resulting $\tau$ matrix, where each cell represents the Kendall correlation between two models. We present only the lower-triangular half of this symmetric matrix for clarity. An adjacent table below the matrix describes each model's type (first row), covariance factor $k_{\text{cov}}$ (second row), and tie factor $k_{\text{tie}}$ (third row). The codes BT, RK, and DV represent Bradley-Terry, Rao-Kupper, and Davidson models, respectively, while BT$^*$ denotes the Bradley-Terry model with ties treated as half win and half loss. Across our 30 models, the Kendall correlation ranged from 0.96 to 1, indicating overall similarity in rankings but with distinguishable differences driven by model

parameter variations. Further insights into these parameter-driven distinctions emerge by reordering the correlation matrix, as detailed in the next section.

### E.4 IDENTIFYING RANKING SIMILARITIES VIA HIERARCHICAL CLUSTERING

To better interpret the distinctions between models, we performed hierarchical agglomerative clustering (Hastie et al., 2009, Section 14.3.12) on the distance matrix $\mathbf{J} - \boldsymbol{\tau}$, where $\mathbf{J}$ is a matrix of all ones, converting $\boldsymbol{\tau}$ into a dissimilarity measure. Using optimal leaf ordering (Bar-Joseph et al., 2001), this clustering reorders the rows and columns of $\boldsymbol{\tau}$ to reveal natural groupings based on ranking similarity across models.

The ordering of models shown in Figure E.3 is directly the arrangement produced by hierarchical clustering, visualized in the dendrogram below the figure, which reveals a distinct block structure in the $\boldsymbol{\tau}$ matrix. The clustering first divides the models into two main groups: those without covariance (indicated by ✗, columns 1 to 10, shown by the yellow branch) and those with covariance (columns 11 to 30). Within the group of models with covariance, further subdivision occurs, with models having $k_{\text{cov}} = 0$ (columns 11 to 20, shown by the orange branch) forming a distinct sub-block from those with $k_{\text{cov}} = 3$ (columns 21 to 30, shown by the red branch). This hierarchical structure highlights that the presence and type of covariance parameter $k_{\text{cov}}$ are primary factors influencing ranking similarity, more so than the tie factor $k_{\text{tie}}$.

While covariance modeling prominently influences ranking consistency, earlier results (see Section 3 and Appendix D) demonstrated that our generalized tie modeling significantly enhances inference and predictive accuracy. This dual impact—covariance structure shaping ranking alignment and generalized tie modeling improving model fit and accuracy—illustrates the complementary strengths of these two generalizations in paired comparison models.

## APPENDIX F   RELATIONSHIP BETWEEN LLM CHARACTERISTICS AND SCORES

This section explores the relationship between the scores derived from our ranking framework and three key characteristics of large language models: the number of parameters, computational budget (FLOPs), and dataset size. Using data from Epoch AI (2022), which provides these attributes for various LLMs, we matched these models with those evaluated in our analysis. To ensure reliability, we filtered the data to include only models with confident values for these characteristics. Among the matched models, 31, 27, and 24 LLMs had available and confident values for the number of parameters, FLOPs, and dataset size, respectively. Figure F.1 presents scatter plots illustrating these relationships, with the abscissa displayed on a logarithmic scale and dashed regression lines capturing the linear trends. The scores, shown on the ordinate, are derived from our generalized Rao-Kupper model corresponding to model 18 of Table D.1.

Table F.1 summarizes the results of independent OLS regressions performed for the logarithm of each characteristic against the scores. The table presents the number of LLMs included in each regression, along with the regression coefficients and their standard errors, $p$-values, coefficient of determination ($R^2$), and Pearson correlations ($r$). The results demonstrate consistent positive associations between all three characteristics and model scores. Computational budget (FLOPs) and dataset size exhibit the strongest associations, with $R^2 = 0.59$ and $r = 0.77$ for FLOPs, and $R^2 = 0.50$ and $r = 0.70$ for dataset size. The number of parameters shows a weaker association, reflected by $R^2 = 0.15$ and $r = 0.38$, though it remains statistically significant.

Table F.1: Regression results examining the relationship between LLM characteristics and scores.

| Variable | # LLMs | Coeff ($\times 100$) | $p$-Value | $R^2$ | Pearson $r$ |
|---|---|---|---|---|---|
| Number of Parameters | 31 | 1.76 ($\pm 0.79$) | 3.4144% | 0.15 | 0.38 |
| Training Compute (FLOPs) | 27 | 2.98 ($\pm 0.50$) | 0.0003% | 0.59 | 0.77 |
| Dataset Size | 24 | 5.70 ($\pm 1.23$) | 0.0124% | 0.50 | 0.70 |

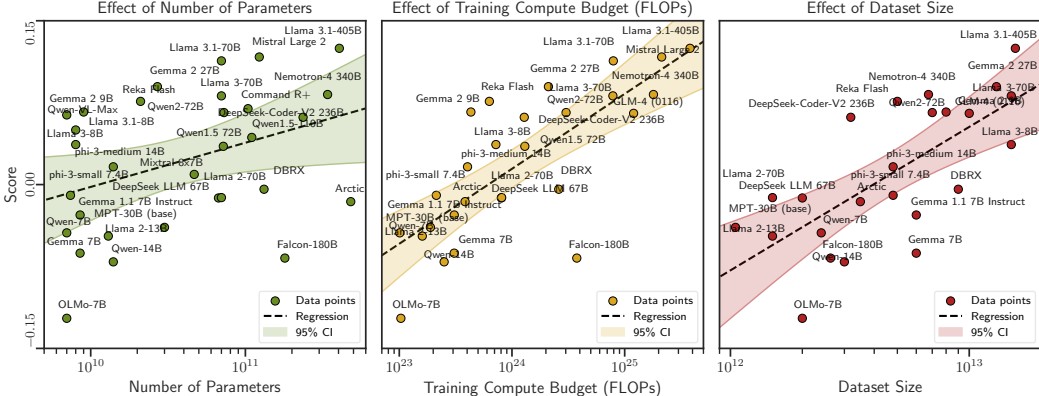

Figure F.1: Scatter plots showing the effect of LLM characteristics on scores. The abscissa represents the logarithmic scale of the characteristics: (left) number of parameters, (middle) computational budget (FLOPs), and (right) dataset size. The ordinate (y-axis) is shared across all panels and shown only for the left panel. Each point is labeled with the corresponding LLM name, and the regression lines are shown in dashed black.

Our findings align with prior studies on the scaling laws of LLMs. Kaplan et al. (2020) emphasize that model performance depends strongly on scale, encompassing model size, dataset size, and computational budget. Hoffmann et al. (2022) further highlight the importance of balancing these factors, demonstrating that compute and dataset size play pivotal roles in maximizing performance within fixed budgets. Consistently, our analysis shows that scores improve across all three characteristics, with particularly strong trends observed for computational budget and dataset size.

The comparatively modest association observed for the number of parameters in our analysis reflects the diminishing returns noted in the literature when model size is increased without proportional scaling of compute and data resources. This observation underscores the importance of balanced scaling for optimal performance, as emphasized by prior studies.

Given the sparsity of available data for proprietary models in our analysis, these conclusions should be interpreted with caution. Further studies incorporating more comprehensive datasets could provide additional insights into the interplay between computational budget, dataset size, and the number of parameters in driving LLM performance.

## APPENDIX G    IMPLEMENTATION AND REPRODUCIBILITY GUIDE

We developed a Python package `leaderbot`[3] that implements the methods presented in this paper. The package allows users to reproduce the numerical results, evaluate model fit, and explore model generalization performance. Below, we provide examples of using `leaderbot` for common tasks such as model training, evaluation, and visualization. The full documentation, including further functionality and customization options, is available online.

### G.1    MODEL TRAINING AND VISUALIZATION

Listing G.1 demonstrates the basic usage of `leaderbot` for training a statistical model and visualizing results. In this example, we replicate Model 23 from Table D.1 using the `Davidson` class, which includes both tie modeling and covariance. The model is instantiated with parameters $k_{\text{cov}} = 0$ for covariance and $k_{\text{tie}} = 0$ for the tie factor model. Once instantiated, the model is trained using the BFGS optimization method on the dataset. While `leaderbot` ships with the dataset used in

---

[3]`leaderbot` is available for installation from PyPI at https://pypi.org/project/leaderbot. Documentation and usage instructions can be found at https://leaderbot.org. The source code is available on GitHub at https://github.com/suquark/leaderbot.

this paper, the `load` function also allows users to provide a URL to load an external dataset. See documentation for details.

After training, users can perform inference and prediction, compute the loss function and Jacobian at the optimal parameters or any specified parameter values, and generate leaderboard tables. The package also includes visualization tools to replicate figures such as the match matrix (Figure 1), kernel-PCA and MDS plots (Figures 2 and C.1), and hierarchical clustering (Figure C.2), among others.

Listing G.1: Basic usage of `leaderbot` for model training and visualization of results.

```python
# Install leaderbot with "pip install leaderbot"
import leaderbot as lb

# Load the default dataset shipped with the package
data = lb.data.load()

# Create a Davidson model with covariance factor k_cov = 0 (diagonal covariance)
# and tie factor k_tie = 0. This corresponds to Model 23 in Table D.1
model = lb.models.Davidson(data, k_cov=0, k_tie=0)

# Train the model
model.train(method='BFGS', max_iter=1500, tol=1e-8)

# Make inference and prediction
probabilities = model.infer(data)
prediction = model.predict(data)

# Compute the loss function −ℓ(θ), its Jacobian −∂ℓ(θ)/∂θ, and Hessian −∇_θ∇_θᵀℓ(θ)
loss, jac = model.loss(return_jac=True)
hess = model.fisher()

# Plot marginal probabilities (similar to Figure D.1)
model.marginal_outcomes()

# Generate a plot for competitor scores (similar to Figure E.1)
model.plot_scores(max_rank=50)

# Rank competitors based on their scores, print leaderboard
rank = model.rank()
model.leaderboard()

# Visualize correlation using Kernel PCA projected in 3D space (similar to Figure 2)
# Use method='mds' to generate an MDS plot (similar to Figure C.1)
model.map_distance(max_rank=40, method='kpca', dim='3d')

# Generate a match matrix plot for observed and predicted win/loss probabilities
# and tie probabilities (similar to Figure 1)
model.match_matrix(max_rank=25, win_range=[0.2, 0.6], tie_range=[0.15, 0.4])

# Perform hierarchical clustering for the top 100 competitors (similar to Figure C.2)
model.cluster(max_rank=100)
```

## G.2   MODEL EVALUATION: FIT AND CONSISTENCY METRICS

Listing G.2 demonstrates the evaluation of model fit and consistency for five selected models, chosen from the broader set of 30 models discussed in Table D.1, including the original Bradley-Terry model as well as original and generalized versions of the Rao-Kupper and Davidson models.

In this example, each model is trained on the full dataset to evaluate goodness of fit. The script produces a bump chart similar to Figure E.2, comparing the rankings generated by the five models. Additionally, it provides tables similar to Table D.1 and Table D.2, displaying model selection and

goodness-of-fit metrics. These metrics enable users to analyze and compare model consistency in training performance.

Listing G.2: Evaluating model fit and consistency metrics across models in `leaderbot`.

```python
import leaderbot as lb

# Load dataset
data = lb.data.load()

# List models 1, 2, 11, 12, 23, and 24 from Table D.1
models = [
    lb.models.BradleyTerry(data, k_cov=None),
    lb.models.BradleyTerry(data, k_cov=0),
    lb.models.RaoKupper(data, k_cov=0, k_tie=0),
    lb.models.RaoKupper(data, k_cov=0, k_tie=1),
    lb.models.Davidson(data, k_cov=0, k_tie=0),
    lb.models.Davidson(data, k_cov=0, k_tie=1)]

# Pre-train the models
for model in models: model.train()

# Compare model rankings, generating a bump chart like Figure E.2
lb.evaluate.compare_ranks(models, rank_range=[1, 60])

# Evaluate model-selection metrics, similar to Table D.1
mod_metrics = lb.evaluate.model_selection(models, report=True)

# Evaluate models for goodness of fit, similar to Table D.2
gof_metrics = lb.evaluate.goodness_of_fit(models, metric='RMSE', report=True)
```

### G.3 MODEL GENERALIZATION: PERFORMANCE ON TEST DATA

Listing G.3 demonstrates the evaluation of model generalization using a 90/10 train-test split, where the same five models from the previous listing are trained on 90% of the data and tested on the remaining 10%. The resulting RMSE, KLD, and JSD metrics, displayed in a table similar to Table D.3, offer insight into each model's predictive accuracy and robustness on unseen data.

Listing G.3: Evaluating model generalization using train-test split in `leaderbot`.

```python
import leaderbot as lb

# Load dataset
data = lb.data.load()

# Split data into training and test sets
training_data, test_data = lb.data.split(data, test_ratio=0.1, seed=20)

# List models 1, 2, 11, 12, 23, and 24 from Table D.1
models = [
    lb.models.BradleyTerry(training_data, k_cov=None),
    lb.models.BradleyTerry(training_data, k_cov=0),
    lb.models.RaoKupper(training_data, k_cov=0, k_tie=0),
    lb.models.RaoKupper(training_data, k_cov=0, k_tie=1),
    lb.models.Davidson(training_data, k_cov=0, k_tie=0),
    lb.models.Davidson(training_data, k_cov=0, k_tie=1)]

# Evaluate models for generalization on test data, similar to Table D.3
gen_metrics = lb.evaluate.generalization(models, test_data=test_data,
                                         train=True, metric='RMSE', report=True)
```

