# OpenReview forum: "A Statistical Framework for Ranking LLM-based Chatbots"
_ICLR.cc/2025/Conference — ICLR 2025 Poster_

### Official Review · Reviewer_cyAq · 2024-10-31

**Soundness:** 4
**Presentation:** 2
**Contribution:** 3
**Rating:** 6
**Confidence:** 3

**Summary:**

This paper addresses a novel problem in the evaluation of large language model (LLM)-based chatbots by proposing an advanced statistical framework that builds on existing methods used in the Chatbot Arena setting. The framework integrates well-established models, such as Rao-Kupper and Davidson, to account for ties in a rigorous axiomatic framework, and introduces Thurstonian representations to model covariance structures between competitors. These additions allow for more nuanced insights into chatbot rankings and performance consistency. A Python package, leaderbot, is also provided for reproducibility and ease of experimentation.

**Strengths:**

1. **Pioneering Approach**: This paper tackles a novel and important problem in LLM evaluation, presenting a unique perspective on using advanced statistical models to refine the ranking process in chatbot comparisons. Its approach to systematically modeling ties and latent structures sets a new precedent for evaluating LLM-based chatbots and offers fresh insights that go beyond traditional ranking methods.
2. **Innovative Use of Thurstonian Representations**: By integrating Thurstonian models and introducing covariance structures, the paper offers a groundbreaking method for capturing relationships among LLMs. This enables a deeper analysis of model consistency and performance that extends beyond simple rankings, marking a notable advancement in chatbot evaluation techniques.

**Weaknesses:**

1. While the paper introduces an alternative framework, it does not clearly discuss why Arena’s Elo-based approach is insufficient beyond the issue of ties. Additional insight into Arena’s limitations in capturing competitive dynamics or certain statistical shortcomings would strengthen the argument for this new model.
2. The authors mention consistency in high-ranking models and variability in lower-ranking models, but do not explore further distinctions within these groups. For example, identifying specific characteristics (e.g., model size) associated with high consistency could provide more actionable insights.

**Questions:**

1. The paper mentions that high-ranking models show greater consistency than lower-ranked ones. Could you expand on this observation? For example, are high-ranking models generally larger or more sophisticated in architecture? And at what ranking position does this shift in consistency typically occur?
2. How is the effectiveness of handling ties quantified across different statistical models? Is there an analysis of which models are more favorable to certain LLMs, and if so, what might explain these differences?

If these questions can be addressed, the paper could potentially be rated one point higher.

---

> ### Comment · Reviewer_cyAq · 2024-11-26
>
> If I do not receive a response to my questions, I may have to consider lowering the score for the paper. I hope to hear from you soon.

---

> ### Author Response · Authors · 2024-11-26
>
> We appreciate your thoughtful feedback, which has helped us refine and strengthen the manuscript. Below, we address each of your comments in detail:
>
> ---
>
> ## Variability in Rankings and Relationship with LLM Characteristics
>
> The observed stability of rankings for high-scoring LLMs and variability among lower-ranked ones appears predominantly data-driven rather than tied to the intrinsic characteristics of the LLMs themselves. In the Chatbot Arena framework, high-ranking models benefit from substantially more pairwise comparisons, with the top 50 LLMs receiving one to two orders of magnitude more matches than lower-ranked competitors. This abundance of data simplifies the ranking task for statistical models, fostering agreement among different methods.
>
> High-ranked LLMs consistently dominate in match outcomes, and the availability of sufficient data ensures that statistical models with differing assumptions converge to similar rankings. In contrast, lower-ranked LLMs, which are paired less frequently, face limited data availability, leading to greater variability in their inferred rankings. This sparsity makes the rankings of these models more sensitive to the assumptions of specific statistical methods, allowing for greater model-specific interpretations. Models with more parameters (e.g., those incorporating tie modeling and Thurstonian covariance) are better equipped to capture nuanced relationships and uncertainties in pairwise data, thanks to their improved fit to the data.
>
> We also found the reviewer’s suggestion to explore the role of LLM characteristics in relation to ranking insightful. To investigate further, we've added a detailed analysis in **Appendix F**. This section examines the relationship between scores from our framework and three key LLM attributes: number of parameters, computational budget (FLOPs), and dataset size. Using data from Epoch.ai matched with our models, we summarize regression results in a table and provide scatter plots illustrating the log-linear relationships between these characteristics and the scores. Our findings reveal consistent positive associations across all three characteristics, with computational budget and dataset size exhibiting stronger trends, while the number of parameters shows a comparatively weaker association. These observations align with prior studies on scaling laws (e.g., Kaplan et al., 2020; Hoffmann et al., 2022), which emphasize the importance of scaling compute and data alongside model size for optimal performance.
>
> Overall, while variability in rankings is primarily data-driven, this additional analysis highlights intriguing patterns in how LLM characteristics relate to their performance scores.
>
> ---
>
> ## Effectiveness of Handling Ties
>
> Incorporating ties into our statistical models significantly enhances predictive performance, as demonstrated both visually and numerically in our manuscript.
>
> Figure 1 (formerly Figure 2 in the appendix) compares pair-specific win and tie probabilities among 25 competitors. The first column shows empirical data, while subsequent columns depict predictions from various models: our generalized Rao-Kupper model with factored ties (second column), the original Rao-Kupper model with ties (third column), the Bradley-Terry model with ties as half win/loss (fourth column, as used in Chatbot Arena), and the original Bradley-Terry model without ties (fifth column). Notably, our generalized model (second column) aligns closely with the empirical data, capturing both tie and win/loss predictions with *fine-grained accuracy*.
>
> This strong visual alignment is supported by numerical results in Tables D.1, D.2, and D.3, where models incorporating ties consistently achieve improved metrics, including lower negative log-likelihood, cross-entropy, RMSE error, and KL and JS divergences. Additionally, Figure D.1 highlights a significant reduction in prediction error for marginal probabilities (both ties and win/loss outcomes), decreasing by up to two orders of magnitude with tie factor parameters.
>
> Our analysis demonstrates that incorporating ties significantly improves model predictions, enabling better alignment with observed data.
>
> ---
>
> ## Model Favorability to Certain LLMs
>
> We did not conduct an analysis to determine whether specific models are more favorable to certain LLMs. Addressing this would require detailed information about LLM characteristics, such as architecture or training data, which falls outside the current scope of our work, focused on developing a purely statistical framework. However, we see this as a valuable direction for future research.
>
> ---
>
> We’ve revised the manuscript to incorporate these changes and address your comments. Please let us know if there are any remaining concerns or areas where further clarification would be helpful. We appreciate your feedback and look forward to your thoughts.

---

> > ### Comment · Reviewer_cyAq · 2024-11-29
> >
> > I have reviewed your feedback and have decided to maintain my initial scores for now. Thank you for your input.

---

> > > ### Author Response · Authors · 2024-12-03
> > >
> > > Thank you for taking the time to review our responses to your earlier comments. We noticed your recent feedback, where you mentioned maintaining your score "for now." In light of this, we’ve revisited your concerns and prepared additional clarifications and evidence that we believe directly address the points raised. We hope these updates provide the necessary clarity and insights to resolve any remaining questions.
> > >
> > > Should these updates satisfactorily address your concerns, we would greatly appreciate your consideration in revisiting your evaluation.
> > >
> > > ---
> > >
> > > > The paper mentions that high-ranking models show greater consistency than lower-ranked ones. Could you expand on this observation? ... And at what ranking position does this shift in consistency typically occur?
> > >
> > > To further support our earlier response, we generated a new visual analysis. The plot is available in our package repository (anonymized link provided on page 31 of the manuscript) under `/notebooks/counts.pdf`. If permitted, the direct anonymized link to the file is [https://anonymous.4open.science/r/leaderbot-CA90/notebooks/counts.pdf](https://anonymous.4open.science/r/leaderbot-CA90/notebooks/counts.pdf).
> > >
> > > The plot demonstrates:
> > > - A pairwise match frequency matrix showing the frequency of matches between competitors.
> > > - Marginal frequencies of matches for each competitor, highlighting a sharp decline in matches starting at rank 50.
> > >
> > > The match frequency heatmap reveals that top-ranked models (ranks 1–50) compete significantly more among themselves compared to lower-ranked models. The marginal plot further confirms a sharp decline in the total number of matches beyond rank 50, where match counts drop by one to two orders of magnitude. This aligns with the observed shift in ranking consistency: rankings are stable for the top 50 models due to their higher match frequency but become more variable beyond rank 50. This supports our hypothesis that match frequency, not intrinsic characteristics, drives ranking stability.
> > >
> > > ---
> > >
> > > > ... are high-ranking models generally larger or more sophisticated in architecture?
> > >
> > > As described in Appendix F, our analysis shows that higher-ranked models correlate strongly with larger datasets, higher computational budgets, and more parameters, suggesting that these models are generally more sophisticated. While architecture-specific details are unavailable for many of the models, the number of parameters serves as a meaningful proxy, providing insights into the relationship between sophistication and ranking.

---

> > > > ### Author Response · Authors · 2024-12-03
> > > >
> > > > Continued from the previous comment…
> > > >
> > > > > Is there an analysis of which models are more favorable to certain LLMs, and if so, what might explain these differences?
> > > >
> > > > We extended our analysis to address this question. Using the rankings produced by all 30 statistical models, we analyzed the mean rank ($y$) and the variability in rankings ($\delta y$), measured as the standard deviation of ranks across models. To explore potential drivers of this variability, we conducted univariate and multivariate regressions of $\delta y$ as the dependent variable, with mean rank ($y$) and LLM characteristics (number of parameters $x_1$, FLOPs $x_2$, dataset size $x_3$) as predictors. Key findings include:
> > > >
> > > > 1. **Variability and Rank**: Ranking variability ($\delta y$) strongly correlates with mean rank ($y$). Models with higher ranks (lower rank numbers) tend to exhibit less variability in their rankings, while lower-ranked models show larger $\delta y$ values (more variability). This supports our hypothesis that higher match frequencies for top-ranked models stabilize their rankings. Regression results confirm this, with $y$ (mean rank) exhibiting the strongest statistical association with $\delta y$ (p-value = 0.010 in univariate and 0.004 in multivariate models), explaining approximately 28% of the variance in $\delta y$ ($R^2 = 0.28$).
> > > >
> > > > 2. **Limited Impact of Characteristics**: Adding LLM characteristics (number of parameters, FLOPs, dataset size) to the regression provides minimal improvement in explanatory power. Among these, only computational budget (FLOPs) shows an association with $\delta y$ (p-value = 0.01), while the number of parameters and dataset size are insignificant (p-values = 0.2 and 0.17). This reinforces that rank variability is driven primarily by match frequency and data availability, rather than intrinsic LLM characteristics.
> > > >
> > > > 3. **Visual Evidence**: To further illustrate these findings, we created a new plot, which can be viewed here: [https://anonymous.4open.science/r/leaderbot-CA90/notebooks/epoch_ai_var.pdf](https://anonymous.4open.science/r/leaderbot-CA90/notebooks/epoch_ai_var.pdf).
> > > >
> > > >    This plot is similar to Figure F.1 but with key differences: (1) the y-axis shows mean rank ($y$) instead of scores, and (2) the size of scatter points reflects $\delta y$ (rank variability), with larger circles indicating higher variability. The plot highlights that larger circles (higher $\delta y$) are predominantly found among lower-ranked models, while higher-ranked models have smaller circles, indicating less variability. Additionally, the visual trends along the x-axis (LLM characteristics) are a byproduct of their strong correlation with rank, not a causal relationship with rank variability.
> > > >
> > > > While investigating favoritism of statistical models toward specific LLM "types" is a valuable direction, such an analysis requires broader data on LLM characteristics, which are unavailable for most models. These limitations preclude a definitive analysis in this context, but we believe the insights provided here address the reviewer’s concerns meaningfully.

---

> > > > > ### Comment · Reviewer_cyAq · 2024-12-03
> > > > >
> > > > > Thank you for your detailed response. While the initial draft had some flaws in presentation and experimental setup, I found the paper interesting and decided to rate it 6 to encourage you to address my concerns. Although some issues were not fully resolved in the previous version, the latest revisions address most of my points. As a result,  I will keep my rating unchanged which is marginally above the acceptance threshold.

---

### Official Review · Reviewer_FnUn · 2024-11-01

**Soundness:** 3
**Presentation:** 2
**Contribution:** 3
**Rating:** 6
**Confidence:** 4

**Summary:**

The authors propose an improved statistical model for ranking large language models (LLMs) on the Chatbot Arena dataset to enhance the traditional Elo rating system. Paper points out that the Elo system has limitations in handling ties and capturing relationships between models. To address these issues, paper introduces the Rao & Kupper and Davidson models, as well as a novel factor model to better capture the complexity of ties between different models, thereby improving prediction performance. Finally, they provide a Python package, “Leaderbot,” to reproduce the statistical model and support further experiments.

**Strengths:**

1. The authors identify symmetry issues in the likelihood function of traditional models, which could lead to instability in parameter estimation. To address this, they propose symmetry constraints that ensure stable parameter estimation, thereby enhancing the model’s optimization performance and interpretability;
2. They effectively address the issue of ties, which is a limitation of the existing Elo system, and make optimizations to handle this;
3. They provide a Python package that allows for the reproduction of the paper's results and supports further research.

**Weaknesses:**

1. The authors use multiple models to rank models, showing that high-ranking models exhibit greater consistency than lower-ranking ones. However, they do not provide further analysis, such as examining the specific characteristics of models that initially show ranking inconsistencies;
2. The authors’ work focuses on optimizing the ranking model but lacks subsequent analysis. Additional insights, such as a more in-depth examination of correlations between LLMs or an analysis of the differences between Leaderbot rankings and Chatbot Arena Elo rankings in relation to model characteristics, would enhance this work.

**Questions:**

1. In the treatment of pairs and unique pairs, have the authors considered introducing semantic clustering to their model, as opposed to solely using statistical techniques? Additionally, could they clarify the reasoning behind not exploring semantic relationships within pairs?
2. Have the authors considered methods other than PCA to further analyze the correlations between models in the results? The current approach does not sufficiently reveal the underlying correlations between LLMs in an intuitive way.

---

> ### Author Response · Authors · 2024-11-26
>
> We appreciate your thoughtful suggestions—they’ve helped us improve the manuscript significantly. Below, we’ve addressed your comments by expanding our analyses and providing additional insights.
>
> ---
>
> ## Examination of Ranking Consistencies and Model Characteristics
>
> To address your suggestion regarding ranking consistencies and their relation to model characteristics, we’ve added **Appendix Section F: Relationship Between LLM Characteristics and Scores**. This section explores the correlation between performance scores and key LLM attributes, such as the number of parameters, computational budget (FLOPs), and dataset size. Our analysis reveals strong correlations between scores and both computational budget and dataset size, offering insights into the factors driving model performance variability. These results align with established trends in scaling laws and provide actionable insights into the observed consistency of high-ranking models.
>
> ---
>
> ## Analysis of Ranking Variability Across Models
>
> In response to your feedback on examining ranking differences across models more deeply, we expanded the comparative analysis in **Appendix Section E: Comparative Analysis of Ranking Variability**, which now includes:
>
> - **Quantifying Ranking Similarity Across Models (Section E.3):** We systematically quantified ranking similarities across all 30 models using Kendall’s ranking correlation. Notably, *Model 1* in our analysis corresponds to the Elo-based ranking algorithm employed by Chatbot Arena. This provides a direct comparison between Leaderbot rankings and Arena’s Elo rankings, offering a more rigorous analysis beyond the visual bump chart presented in the original submission.
>
> - **Identifying Ranking Similarities via Hierarchical Clustering (Section E.4):** By clustering the Kendall correlation matrix, we uncovered distinct clusters of ranking behaviors. This analysis reveals a fascinating finding: the inclusion or exclusion of the Thurstonian covariance model is the primary driver of these clusters. These results show how covariance modeling shapes ranking consistency, complementing earlier findings on the impact of generalized tie modeling on inference accuracy.
>
> ---
>
> ## Incorporation of Complementary Methods for Correlation Analysis
>
> In response to your comment about exploring methods beyond PCA, we’ve expanded our correlation analysis in **Appendix Section C**. Specifically:
>
> - **Multidimensional Scaling (Section C.2):** We introduced MDS as a complementary method to kernel PCA. MDS provides an intuitive spatial representation of chatbot relationships and, interestingly, uncovers ranking orders solely from score differences and covariance-derived uncertainties, without access to absolute scores.
>
> - **Hierarchical Clustering (Section C.3):** We extended the correlation analysis with hierarchical clustering to identify performance tiers among competitors. This approach, driven by the optimal leaf ordering of the dissimilarity matrix, offers an intuitive grouping of competitors into tiers. Notably, the method recovered the rankings almost perfectly without direct access to the scores, relying on the dissimilarity matrix derived from our generalized models incorporating Thurstonian covariance. This integration of performance scores and covariance-based dissimilarities underscores the interpretability of the clustering approach.
>
> ---
>
> ## Clarifying the Use of Semantic Clustering
>
> Regarding the use of semantic clustering, we believe that statistical clustering techniques, such as hierarchical clustering, are well-suited for our pairwise comparison framework. Our focus was on extracting performance-based groupings and relationships that reflect the pairwise comparisons and score-based dissimilarities of the models. Exploring semantic relationships is an interesting direction, but it would require additional datasets or features, such as linguistic embeddings, which fall outside the scope of this work. To address this point, we’ve added a clarification at the end of *Appendix Section C.3* to highlight this reasoning and emphasize the alignment of our approach with the statistical focus of the manuscript.
>
> ---
>
> We’ve revised the manuscript to reflect these changes and address your feedback. Please let us know if there are any remaining concerns or areas where further clarification would be helpful. We’re grateful for your suggestions and look forward to your thoughts.

---

> > ### Comment · Reviewer_FnUn · 2024-11-30
> >
> > Thank you for your reply, it has addressed some of my concerns, and I will maintain my score.
> >
> > Additionally, I have a small question that doesn't affect my score. In Appendix Section F, Figure F.1, you have included the Yi-Large, a closed-source model with unpublished details. Could you please explain how this was achieved?

---

> ### Author Response · Authors · 2024-12-02
>
> Thank you for your follow-up question regarding Yi-Large in Figure F.1. Following your observation, we reviewed the source data from epoch.ai and identified that the values for Yi-Large, along with a few other models, were derived from external estimates rather than directly published details. In response, we refined our methodology in Appendix F to include only models flagged as "confident" in epoch.ai's dataset, ensuring the accuracy of our analysis.
>
> This refinement adjusted the number of LLMs in the regression analysis from 37, 34, and 28 to 31, 27, and 24 for the number of parameters, computational budget (FLOPs), and dataset size, respectively. Notably, this adjustment led to *improved* regression results, with stronger $R^2$, $r$, and $p$-values, indicating more reliable associations between these characteristics and the scores. Below is the updated regression Table F.1:
>
> | **Variable**              | **# LLMs** | **Coeff ($\times 100$)** | **$p$-Value**   | **$R^2$**   | **Pearson $r$**   |
> |---------------------------|------------|--------------------------|-----------------|-------------|-------------------|
> | Number of Parameters      | $31$         | $1.76 (\pm 0.79)$     | $3.4144\%$        | $0.15$        | $0.38$              |
> | Training Compute (FLOPs)  | $27$         | $2.98 (\pm 0.50)$     | $0.0003\%$        | $0.59$        | $0.77$              |
> | Dataset Size              | $24$         | $5.70 (\pm 1.23)$     | $0.0124\%$        | $0.50$        | $0.70$              |
>
> The updated figure F.1 is now available in our package repository (anonymized link provided on page 31 of the manuscript) under `/notebooks/epoch_ai.pdf`. This plot includes the confidence intervals for the regression and reflects the updated analysis. If permitted, the direct anonymized link to the file is: [https://anonymous.4open.science/r/leaderbot-CA90/notebooks/epoch_ai.pdf](https://anonymous.4open.science/r/leaderbot-CA90/notebooks/epoch_ai.pdf). The manuscript has also been updated to reflect the new data in the table and this plot.
>
> We appreciate your attention to this detail, which has ultimately improved our findings.

---

### Official Review · Reviewer_ztF7 · 2024-11-03

**Soundness:** 3
**Presentation:** 1
**Contribution:** 1
**Rating:** 5
**Confidence:** 3

**Summary:**

This paper presents a statistical framework for ranking LLM-based chatbots. It addresses the limitations of existing methods like the Elo rating system by incorporating ties and covariance structures. The proposed apply well-established statistical models to properly account for ties within an axiomatic framework and introducing factor analysis. Experiments on the Chatbot Arena dataset show improved accuracy and insights into chatbot rankings and relationships.

**Strengths:**

1. The proposed method can account for ties in an axiomatic framework. This improves not only tie prediction but also enhances win-loss inference.
2. The mathematical analysis is comprehensive and thorough.
3. This work has an open source python package which is easy to use.

**Weaknesses:**

1. For model ranking, it has no "ground truth". Therefore, it is hard to convince others under this method, the ranking is more accurate.
2. The evaluation is highly dependent on the Chatbot Arena dataset which makes this work a slight improvement on chatbot arena. Therefore, the impact of this work is limited.
3. As for chatbot arena, a simple enough ranking rule is more important if the users are common users. This work will make the rule too complicated for them to understand and then reduce the impact of chatbot arena.

**Questions:**

1. Is this method only useful to chatbot arena or it is useful to all Elo ratings?
2. We need another figure to see the ranking difference of original chatbot arena and this methods. Also the figure 1 showing the result of PCA analysis is confusing.

---

> ### Author Response · Authors · 2024-11-26
>
> We appreciate your feedback and for highlighting strengths as well as areas for improvement/clarification. Below, we address your concerns and clarify points raised.
>
> ---
>
> ## On the Absence of Ground Truth in Ranking
>
> We agree that rankings lack a definitive ground truth, as they are inferences drawn from models and data. However, in statistical modeling, the *goodness of fit* and *predictive performance* of a model can and should be rigorously evaluated against empirical data, which serves as the ground truth for observed outcomes.
>
> To this end, we systematically compared the predictive performance and the goodness of fit of our models against empirical data:
>
> - Figure 1 (formerly Figure 3 in the appendix) demonstrates the predictive performance of our model in capturing tie and win/loss probabilities. Notably, our generalized model achieves *fine-grained accuracy*, closely aligning with empirical data, and outperforms simpler approaches, including the Elo-based method.
> - **Appendix D** evaluates model selection and goodness of fit metrics (Tables D.1, D.2), including negative log-likelihood, cross-entropy, RMSE error, and divergences, demonstrating superior fit of generalized models.
>
> While rankings themselves are an *inference* without a definitive ground truth, the framework underlying these rankings can be rigorously validated based on its ability to model and predict empirical data accurately.
>
> ---
>
> ## Dependence on the Chatbot Arena Dataset and Broader Impact
>
> We acknowledge the reviewer’s observation that our experiments focus on the Chatbot Arena dataset. This pioneering dataset, widely used by OpenAI, Google, and Hugging Face, is among the largest and most credible for pairwise comparisons in LLM evaluation.
>
> While our experiments leverage this dataset, our contributions extend far beyond it. The framework we propose—incorporating ties and covariance—addresses fundamental challenges in paired comparison frameworks, applicable to diverse domains regardless of dataset. To clarify the general applicability of our work, **Appendix A** highlights paired comparison applications across sports analytics, marketing, psychometrics, election studies, and clinical trials.
>
> Additionally, paired comparison methods are increasingly applied in modern machine learning, such as reinforcement learning with human feedback (RLHF) and direct preference optimization (DPO), key to fine-tuning large language models. Recent studies (e.g., Rafailov et al., 2024; Karthik et al., 2024; Liu et al., 2024) showcase the growing relevance of paired comparison models in these settings. Our contributions enhance the flexibility of these models, underscoring their potential to address emerging challenges.
>
> ---
>
> ## Balancing Sophistication with User Simplicity
>
> We understand the concern about balancing complexity with user accessibility. The statistical framework operates behind the scenes and does not affect the simplicity of the user interface. Rankings remain straightforward for users, supported by an intuitive Python package with an easy-to-use interface.
>
> In practice, grounding rankings in a robust statistical framework can enhance user trust in the platform, even if the underlying methodology is sophisticated.
>
> ---
>
> ## Visualization of Ranking Differences
>
> Thank you for suggesting a figure illustrating ranking differences between Chatbot Arena’s method and ours. **Figure E.2** (bump chart), previously included as Figure 5 in the earlier submission, tracks ranking differences across models, including Model 1, representing Chatbot Arena's Elo-based system. To clarify, we updated the figure caption to explicitly label Model 1 as the Chatbot Arena method. We are also open to including additional visualizations if you believe they would add further clarity.
>
> Beyond the bump chart:
> - **Appendix E.3** quantifies ranking similarities using Kendall’s rank correlation across all 30 models, providing a systematic metric for ranking comparison.
> - **Appendix E.4** applies hierarchical clustering to Kendall’s correlation matrix, revealing patterns in ranking behavior and how model features drive similarities.
>
> ---
>
> ## Interpretation of PCA in Figure 2
>
> We acknowledge the challenge of interpreting the 3D PCA scatter plot in **Figure 2** (formerly Figure 1), as overlapping points and labels limit clarity. To address this, **Appendix C.2** introduces a 2D Multidimensional Scaling (MDS) plot, offering similar insights into the covariance structure with improved interpretability. As kernel PCA and MDS are dual methods visualizing the same dissimilarity matrix, both provide consistent insights. This additional visualization aims to enhance clarity while preserving the depth of analysis.
>
> ---
>
> We've revised the manuscript to address your comments. If there’s anything else you’d like us to clarify or discuss further, please let us know—we’d be happy to engage. Thank you again for your feedback and for helping us improve our work.

---

> > ### Comment · Reviewer_ztF7 · 2024-11-28
> >
> > Thanks for your thorough response.
> > I will increase score slightly.

---

### Author Response · Authors · 2024-11-26

Thank you for your constructive feedback. Some comments highlighted areas where our explanations could be clearer, while others raised insightful questions that inspired us to expand our analyses and discussions. These suggestions have significantly enhanced the clarity and presentation of our manuscript. Below, we summarize the key enhancements and additions.

---

## Enhanced Analysis of Ranking Variability Across Models

To provide a more systematic and comprehensive examination of ranking variability, we expanded the analysis in **Appendix Section E: Comparative Analysis of Ranking Variability**, building on the visualization provided by the bump chart in the previous submission. Key additions include:

- **E.3: Quantifying Ranking Similarity Across Models.** Using Kendall's ranking correlation matrix, we systematically quantified the alignment of rankings across all 30 models. This approach offers a robust mathematical framework for comparing ranking similarities, going beyond the visual insights provided by the bump chart.

- **E.4: Identifying Ranking Similarities via Hierarchical Clustering.** By applying hierarchical clustering to the Kendall correlation matrix, we identified distinct clusters of ranking behaviors. This analysis revealed an interesting result: the inclusion or exclusion of the Thurstonian covariance model is the main factor driving these clusters. These findings illustrate how covariance modeling shapes ranking consistency and complements earlier results on the impact of generalized tie modeling on inference accuracy.

---

## Incorporation of Complementary Methods for Correlation Analysis

We expanded our correlation analysis by introducing **multidimensional scaling (MDS)** in **Appendix Section C.2: Interpretation and Visualization of Covariance**. This method complements kernel PCA from the original submission by offering an intuitive spatial representation of chatbot relationships. MDS arranges competitors in a configuration that mirrors their rankings—achieved without direct access to absolute scores. Instead, this structure is derived solely from score differences and covariance-based uncertainties, effectively uncovering the ranking order.

In **C.3: Hierarchical Clustering of Competitor Performance**, we extended this analysis to identify performance tiers among competitors. The hierarchical clustering approach, with optimal leaf ordering, not only grouped competitors into meaningful tiers but also retrieved their ranking order purely from the dissimilarity matrix derived from our generalized model. This result highlights the strength of the dissimilarity matrix in capturing both relative performance and structural relationships among competitors.

---

## Exploration of Model Characteristics and Rankings

In response to comments about the relationship between ranking and model characteristics, we introduced **Appendix Section F: Relationship Between LLM Characteristics and Scores**. This section explores the connection between performance scores and attributes such as the number of parameters, computational budget (FLOPs), and dataset size. Our analysis highlights strong correlations between performance scores and computational budget (FLOPs) and dataset size, consistent with trends in scaling laws.

---

## Clarifying Covariance and Identifiability

We added **Appendix Section B: Unidentifiability of Parameters in Paired Comparison Models** to provide a foundational understanding of the unidentifiability of covariance in pairwise comparison frameworks. This section sets the stage for the subsequent analysis in **Section C**, ensuring that the covariance-based methods employed in our work are well-justified.

---

In addition to these substantive analyses, we made a few adjustments to improve clarity and readability. For instance:
- Table 1 from the previous submission has been moved to **Appendix Section D.1**.
- Figure 2, previously in the appendix, has been moved to the main body as **Figure 1** with slight reordering of its panels for better interpretability.

We hope these revisions address your comments and enhance the overall quality of the manuscript. If there are any remaining concerns or areas where further clarification would be helpful, please let us know. We truly appreciate your feedback and the opportunity to improve our work.

---

### Meta-Review · Area_Chair_SoLr · 2024-12-22

**Metareview:**

This paper proposes an improved version of ELo rating in Chatbot Arena by considering ties and covariances by using Rao & Kupper and Davidson models. Experiments found that the proposed improvement is better at modeling empirical data.

Strengths:
1. The method improves the ELo rating by considering ties and covariances.
2. The authors also released an open-source Python package.

Weaknesses:
1. It is not clear how much benefits we get by considering ties compared to the existing ELo ratings.
2. As reviewers pointed out, the proposed metric is harder to be understood by normal users of Chatbot Arena.

Since reviewers are overall positive about this paper, I'm recommending acceptance, but I wouldn't mind if the paper gets rejected.

**Additional Comments On Reviewer Discussion:**

Most reviewer questions seek clarifications and there's no significant change to the paper during the rebuttal period.

---

### Decision · Program_Chairs · 2025-01-22

Accept (Poster)